# An Influenza A virus can evolve to use human ANP32E through altering polymerase dimerization

Carol M. Sheppard [1,4,5] ✉, Daniel H. Goldhill [1,3,4], Olivia C. Swann[1], Ecco Staller[2], Rebecca Penn[1], Olivia K. Platt [1], Ksenia Sukhova [1], Laury Baillon[1], Rebecca Frise[1], Thomas P. Peacock [1], Ervin Fodor [2] & Wendy S. Barclay [1,5] ✉

Human ANP32A and ANP32B are essential but redundant host factors for influenza virus genome replication. While most influenza viruses cannot replicate in edited human cells lacking both ANP32A and ANP32B, some strains exhibit limited growth. Here, we experimentally evolve such an influenza A virus in these edited cells and unexpectedly, after 2 passages, we observe robust viral growth. We find two mutations in different subunits of the influenza polymerase that enable the mutant virus to use a novel host factor, ANP32E, an alternative family member, which is unable to support the wild type polymerase. Both mutations reside in the symmetric dimer interface between two polymerase complexes and reduce polymerase dimerization. These mutations have previously been identified as adapting influenza viruses to mice. Indeed, the evolved virus gains the ability to use suboptimal mouse ANP32 proteins and becomes more virulent in mice. We identify further mutations in the symmetric dimer interface which we predict allow influenza to adapt to use suboptimal ANP32 proteins through a similar mechanism. Overall, our results suggest a balance between asymmetric and symmetric dimers of influenza virus polymerase that is influenced by the interaction between polymerase and ANP32 host proteins.

Influenza A viruses emerging from wild birds into humans have caused three pandemics since 1918[1]. Influenza A viruses have also transmitted to several other mammalian hosts including pigs, dogs, horses, bats, seals and whales[1]. Like all viruses, influenza A viruses need to co-opt host factors to successfully replicate within a host cell[2]. Influenza A virus requires host factors at many different stages of its life cycle from entry, transport of gene segments to the nucleus, replication, packaging to exit[3–5]. As a result, the virus must adapt to novel host factors to successfully emerge in a new host species[1]. Pandemic preparedness requires us to identify mutations which allow or prevent animal viruses

from using human host factors and to understand the mechanisms that enable host switching[6].

The influenza A virus polymerase is a heterotrimer composed of three subunits PB2, PB1 and PA. The enzyme is highly dependent on the cellular environment and co-opts a large repertoire of host factors to support its activity for transcription and replication of the negative sense RNA genome[2,7]. ANP32 proteins are key host factors that are essential for influenza virus replication[8,9]. They are also important for determining the host range of influenza virus[10]. Birds and mammals each express three members of the ANP32 family; ANP32A, avian/

[1]Department of Infectious Disease, Imperial College London, London, UK. [2]Sir William Dunn School of Pathology, University of Oxford, Oxford, UK. [3]Present address: Department of Pathobiology and Population Sciences, Royal Veterinary College, London, UK. [4]These authors contributed equally: Carol M. Sheppard, Daniel H. Goldhill. [5]These authors jointly supervised this work: Carol M. Sheppard, Wendy S. Barclay. ✉e-mail: carol.sheppard08@imperial.ac.uk; w.barclay@imperial.ac.uk

mammalian ANP32B (paralogues) and ANP32E[11]. All have two domains: a leucine rich repeat (LRR) with a solenoid structure linked to a C-terminal low complexity acidic region (LCAR). The human ANP32 proteins are involved in a multitude of cellular processes including protein trafficking, phosphatase regulation, and apoptosis[12]. They also all have roles associated with chromatin remodelling; huANP32A and huANP32B bind to H3-H4 histones, whereas huANP32E preferentially binds to H2A.Z histones[13,14]. ANP32 proteins are needed by influenza virus polymerase to support replication[15–17]. Mammalian-adapted influenza viruses can use either mammalian ANP32A or ANP32B, with different viruses showing a preference in different species[18,19]. The finding that human cells engineered to lack expression of ANP32A and B do not support influenza A virus polymerase activity nor viral replication suggests that influenza A viruses do not utilize human ANP32E[8,9]. Avian ANP32A contains a 33 amino acid insertion compared to mammalian ANP32 proteins[10] although alternative splicing can result in the formation of shorter avian ANP32As with either a 29 amino acid insertion or no insertion[20,21]. Avian influenza viruses rely solely on ANP32A in avian cells; avian ANP32B and ANP32E show no ability to support influenza A polymerase and chicken cells engineered to lack expression of chANP32A do not support viral polymerase activity nor virus replication[11,22]. When avian influenza A viruses adapt to mammals including humans, adaptive mutations in polymerase such as the PB2 E627K mutation allow the virus to use the shorter form of ANP32A or B that lack the 33 additional amino acids. This is due to enhanced interactions of the 627K polymerase with a stretch of negatively charged amino acids within the LCAR domain of human ANP32 proteins[15,23].

Structural analyses of influenza A polymerases of human and avian viruses reveal that the polymerase complex can form a symmetric dimer[24,25]. The majority of the symmetric dimer interface is formed by PA with smaller contributions from amino acids in PB1 and PB2. The formation of the symmetric dimer has been shown to be required for vRNA synthesis from a cRNA template, with one of the polymerases acting as a replicase and the other as a trans-activating polymerase that enables the realignment of the influenza genome during replication[24,26]. A recent structure of the influenza C polymerase in complex with ANP32A shows that the polymerase can form an alternative asymmetric dimer with the LRR of ANP32 bridging two polymerase enzymes, one bound to the viral promoter RNA and therefore a replicase and the other proposed as an encapsidating polymerase that could load onto the nascent RNA[15]. Recent data suggest that polymerase dimerization is regulated during infection[27,28]. However, the exact mechanism through which this is achieved and the role of ANP32 proteins is not clear.

Host factors have been proposed as targets for new antiviral drugs and could be targets for gene-editing to create farm animals such as poultry or swine resistant or resilient to influenza[11,29]. However, it is not clear whether viruses can evolve to evade antiviral strategies that target host factors, and this has only been studied in a few instances[30,31]. ANP32 proteins are potential targets for novel influenza intervention strategies.

In this study, we investigate whether influenza A virus can evolve to replicate in human cells lacking expression of ANP32A and ANP32B. We describe one example of a virus that shows limited replication in cells lacking human ANP32A and B, and evolves in those cells to increase replication efficiency compared to the ancestor virus. We demonstrate that the mutated influenza polymerase is able to co-opt an alternate host factor, ANP32E. Unexpectedly, the mutations in PB1 and PA, which allow the use of the previously non-functional ANP32E, map to the interface of the symmetric dimer, even though ANP32 interacts with the asymmetric polymerase dimer. Moreover, the adapting mutations reduce symmetric polymerase dimerization, suggesting a link between ANP32 support and dimerization states of influenza virus polymerase.

## Results

### Experimental evolution of an influenza A virus in the absence of human ANP32A and ANP32B

We previously showed that influenza virus polymerase activity and virus replication were generally abrogated in human cells that did not express ANP32A and B (DKO cells). However, we and others have also reported low levels of replication for certain influenza A viruses on these cells[8,9]. A 6:2 reassortant virus (Tky05), showed low infectious titres <10^2 p.f.u/ml on DKO cells after 48 h (Fig. 1A). This virus has an internal gene set derived from a highly pathogenic avian influenza virus, A/turkey/Turkey/05/2005(H5N1), with the mammalian-adapting PB2 mutation E627K[10,32] and external proteins, HA and NA, from PR8[33]. To test whether this virus could evolve to replicate efficiently in the absence of huANP32A and ANP32B, we passaged three independent populations of Tky05 on DKO cells for four passages. Three independent control populations were passaged on WT eHAP cells. After two passages, we found that titres of all three evolved populations in DKO cells were comparable to titres attained by the ancestral virus on WT eHAP cells (Supplemental Fig. 1). Infection of DKO cells with a standard MOI (0.01) of each of the three evolved populations resulted in high titres 48 h post infection whereas infection of DKO cells with the 3 populations that had been passaged in WT eHAP cells did not yield infectious virus. Yields of all 6 populations were similar in WT eHAP cells (Fig. 1B). We sequenced the DKO evolved populations and found mutations present at differing frequencies in segments coding for the polymerase subunits PA, PB1 and PB2, and in one population, an additional mutation in NS (Fig. 1C). We twice plaque-purified and chose a plaque from population 2 for further characterization. Sequencing of this clone revealed just the two mutations in the polymerase subunits: K577E in PB1 and Q556R in PA and no changes in PB2, NS or the PR8 HA and NA segments. We compared the growth of the plaque-purified virus to its unpassaged WT ancestor (Fig. 1D, E). In MDCK cells there was a marginal increase in final titre attained by the mutant K577E + Q556R virus (Fig. 1D). In eHAP WT cells we observed slightly accelerated growth of the K577E + Q556R virus at 8 h post infection but this did not result in an overall growth advantage post 16 h. In DKO cells however the mutant virus replicated productively, and attained titres of 10^6 PFU/ml by 48 h post infection, in contrast with the ancestor, which did not yield virus until 72 h post infection when 10^4 PFU/ml were detected. This late increase in titre for the WT virus was reminiscent of the observation in Fig. 1A, and may represent within-experiment adaptation.

### PB1 K577E and PA Q556R enable polymerase activity in the absence of ANP32A/B

To investigate the effect of PB1 K577E and PA Q556R on influenza polymerase activity in DKO cells, we introduced these mutations into the respective expression plasmids to reconstitute the heterotrimeric polymerase. A minigenome assay was performed in the presence or absence of exogenous huANP32B (Fig. 2A). As shown previously, Tky05 polymerase was not active in DKO cells but activity could be rescued by the introduction of exogenous huANP32B. In contrast, the reconstituted mutant polymerase, K577E + Q556R bearing both PB1 and PA mutations, displayed significant activity in the DKO cells. Furthermore, the expression of huANP32B increased the activity of the K577E + Q556R polymerase to 20 times that of WT. To investigate the contribution of each of the mutations, we assessed them individually in the minigenome assay. Neither the PB1 K577E polymerase nor the PA Q556R polymerase alone displayed significant polymerase activity in DKO cells, although PB1 K577E activity was detectable and above WT. Upon the addition of huANP32B, the PB1 K577E polymerase showed very high activity whereas PA Q556R polymerase showed activity comparable to WT. Thus, both K577E and Q556R were necessary for robust polymerase activity in DKO cells but K577E was solely responsible for the increased activity in the presence of huANP32B. Western

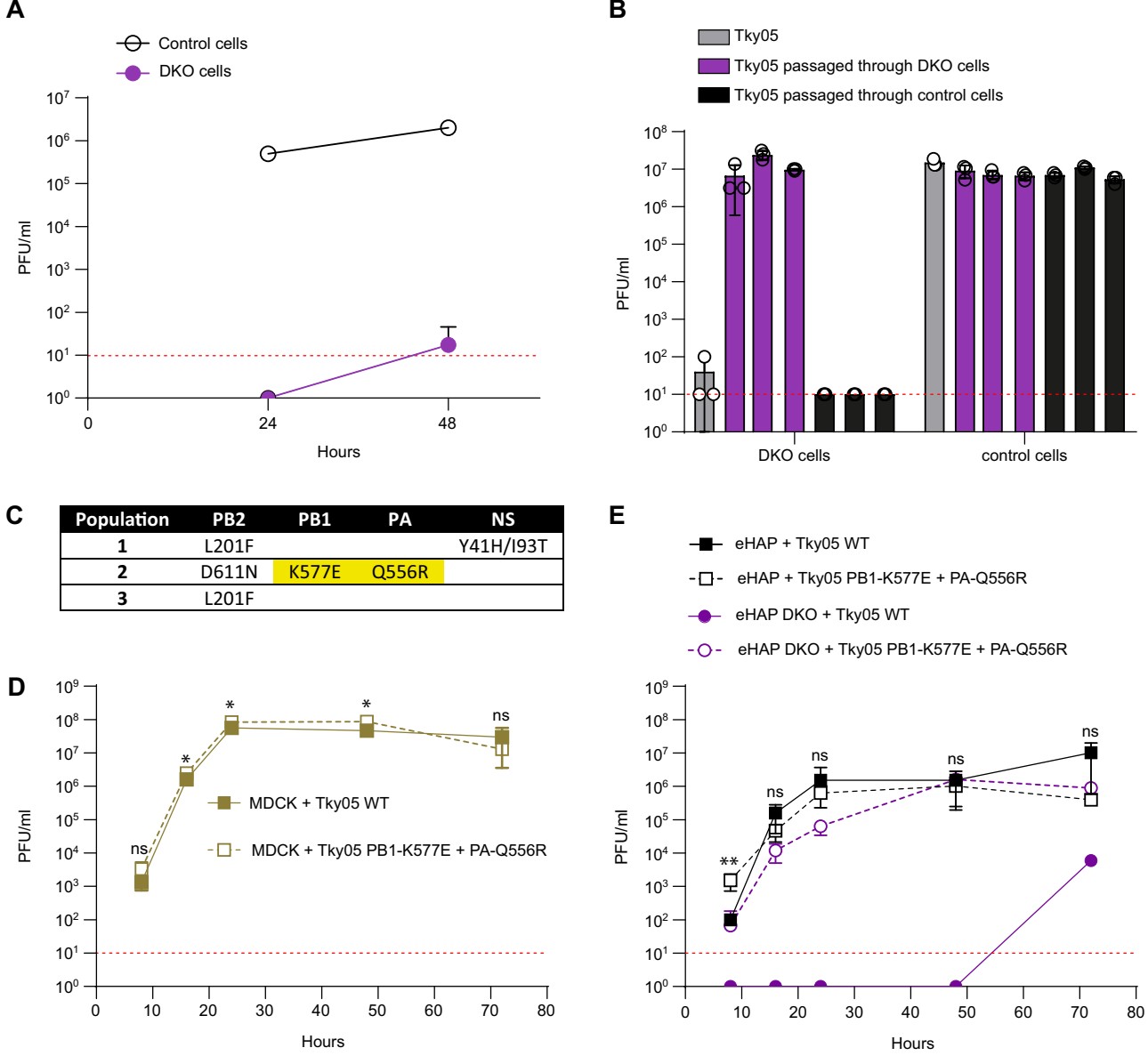

**Fig. 1 | An influenza A virus evolved to grow in human cells lacking ANP32A and ANP32B proteins. A** Tky05 was inoculated on 12-well plates of eHAP (control) and DKO cells at an MOI of 0.0005. Samples were titred on MDCK cells via plaque assay at 24 and 48 h post infection. Data are presented as mean values +/− SD. **B** Following two passages through either DKO cells or WT eHAP cells (control), 1000 PFU from each population as well as the ancestor (Tky05) were inoculated onto six-well plates of DKO and eHAP cells. After 48 h, viruses were titred on MDCK cells via plaque assay. $n = 3$ wells, data are presented as mean values +/− SD. **C** Mutations from populations passaged through DKO cells were discovered using Sanger sequencing. Highlighted mutations were found in the plaque-purified virus, which was

used for future experiments. Growth curves of Tky05 compared to plaque-purified virus containing PB1 K577E, PA Q556R on MDCK (**D**) or eHAP (**E**) cells. 12-well plates were infected at MOI 0.001 (**D**) or 0.0005 (**E**), and viral titres were measured on MDCK cells at 8, 16, 24, 48 and 72 h. $n = 3$ wells, data are presented as mean values +/− SD. Statistical significance was determined by multiple $t$-test of log-transformed data, $*P < 0.05$, $**P < 0.01$, $***P < 0.001$, $****P < 0.0001$, (1D; 8 hpi $p = 0.238$, 16 hpi $p = 0.206$, 24 hpi $p = 0.035$, 48 hpi $p = 0.017$, 72 hpi $p = 0.339$. 1E; 8 hpi $p = 0.003$, 16 hpi $p = 0.101$, 24 hpi $p = 0.802$, 48 hpi $p = 0.600$, 72 hpi $p = 0.066$) Source data are provided as a Source Data file.

blots of polymerase protein expression revealed that the K577E mutation led to enhanced stability of PB1 (Fig. 2A). To test the effect of the PB1 expression levels on polymerase activity we conducted a minigenome assay with a titration of PB1 WT (Supplemental Fig. 2). At levels comparable to PB1 K577E, WT PB1 did not increase polymerase activity, thus higher PB1 K577E protein levels are not responsible for activity in DKO cells.

## PB1 K577E and PA Q556R enable polymerase activity to be supported by ANP32E

The ability of the K577E + Q556R polymerase to function in DKO cells could be due to the polymerase evolving to no longer be dependent

on ANP32 proteins as host factors or due to the polymerase evolving to use a novel host factor in place of huANP32A/B. We hypothesized that, in the latter scenario, the most likely novel host factor would be huANP32E, as it is the most closely related protein to huANP32A and huANP32B (58% similarity)[1]. To test whether ANP32E could act as a replacement for huANP32A/B, we compared the effect of titrating huANP32B, huANP32E or chANP32B in a mini-genome assay performed in DKO cells (Fig. 2B). We have previously shown that chANP32B is unable to support WT avian influenza A virus polymerase function and others have shown that huANP32E is also inactive for influenza A[11,22]. As before, exogenously expressed huANP32B stimulated activity of the K577E + Q556R polymerase to

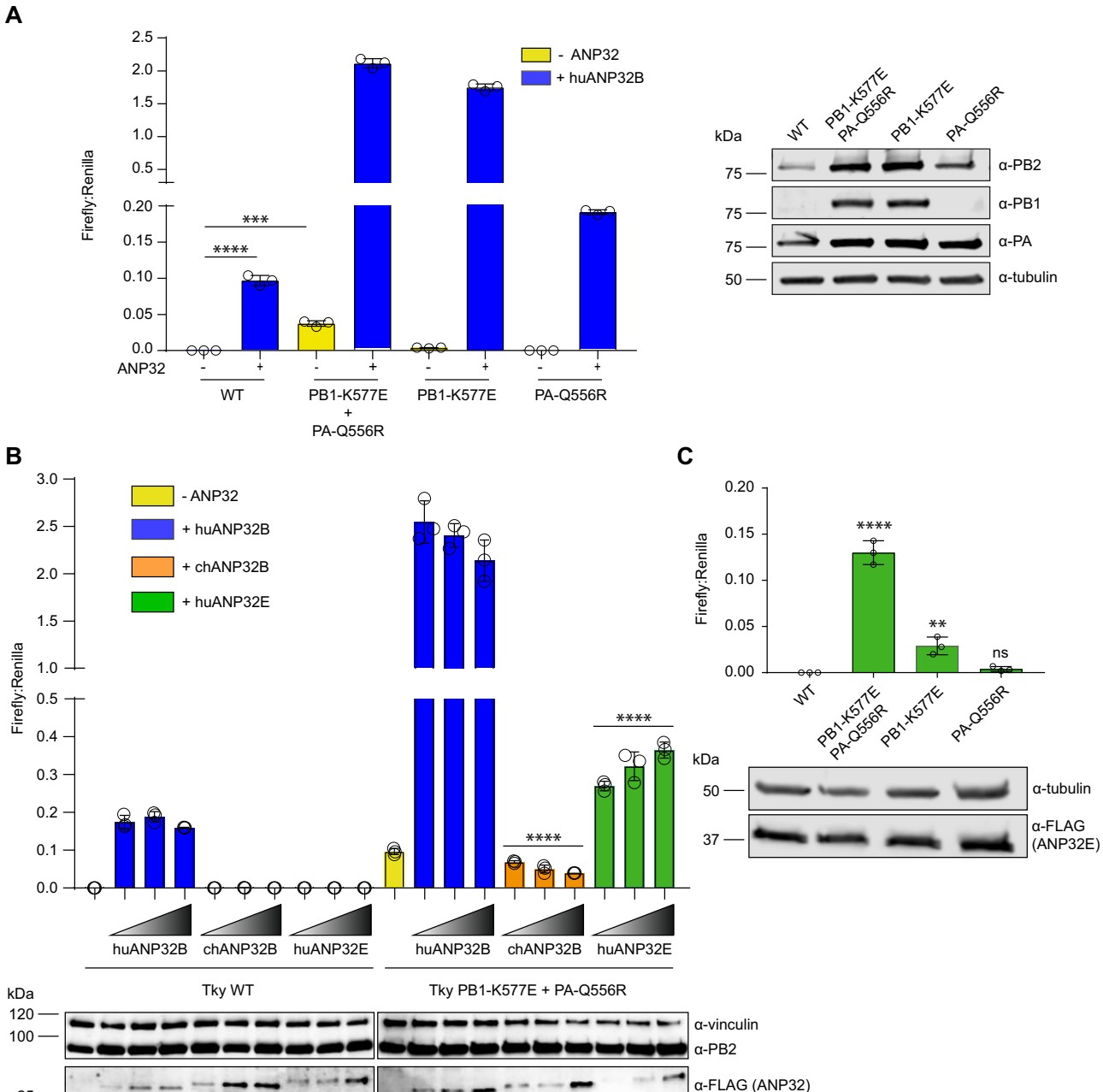

**Fig. 2 | Tky05 PB1 K577E and PA Q556R mutations enhance influenza polymerase activity in the absence of human ANP32A/B.** Minigenome assays were performed in 24-well plates of eHAP DKO cells transfected with pCAGGS Tky05 PB2 (0.04 μg), PB1 or K577E (0.04 μg), PA or Q556R (0.02 μg), NP (0.08 μg), reporter pPolI-luc (0.08 μg) and control pCAGGS-*Renilla* luciferase (0.04 μg) and (**A**) +/− huANP32B-FLAG (0.04 μg). Western blot showing expression of PB2, PB1, PA and tubulin. **B** Minigenome with 0, 0.02, 0.04 or 0.08 μg of huANP32B, chANP32B and huANP32E. Accompanying Western blot showing expression of vinculin, PB2 and ANP32-FLAG. **C** Minigenome with 0.08 μg of huANP32E. All data presented is representative of $n = 3$ biological repeats each conducted with $n = 3$ wells, presented as mean values +/− SD. Statistical significance was determined by multiple comparisons of a one-way ANOVA (**A**, **C**) or test for trend of a one-way ANOVA (**B**) *$P < 0.05$, **$P < 0.01$, ***$P < 0.001$, ****$P < 0.0001$ (**A**; Tky WT -ANP32 vs Tky WT +huANP32B $P < 0.0001$, Tky WT -ANP32 vs Tky PB1-K577E + PA-Q556R -ANP32 $p = 0.00015$, **B**; trend for Tky PB1-K577E + PA-Q556R chANP32B titration $P < 0.0001$, trend for Tky PB1-K577E + PA-Q556R huANP32E titration $P < 0.0001$, **C**; Tky WT vs Tky PB1-K577E + PA-Q556R $p < 0.0001$, Tky WT vs PB1-K577E $p = 0.0063$, Tky WT vs PA-Q556R $p = 0.878$). Source data are provided as a Source Data file.

20 times more than of the WT. In both cases, the lowest concentration of huANP32B was already at saturating levels with no further increase in polymerase activity observed with increasing protein expression. In contrast, chANP32B provided no support for the WT polymerase and exhibited a slight but significant inhibitory effect on the K577E + Q556R polymerase as protein concentrations increased. Remarkably, huANP32E demonstrated significant levels of enhancement of K577E + Q556R polymerase activity in a dose-dependent manner. Conversely, huANP32E did not rescue activity of

WT polymerase in DKO cells despite equivalent levels of protein expression as for huANP32B.

We next tested which of the individual mutations were more affected by the addition of huANP23E. Addition of excess huANP32E to DKO cells conferred appreciable activity to the K577E polymerase and slight but non-significant activity with Q556R (Fig. 2C). K577E and Q556R combined synergistically to give increased polymerase activity with huANP32E greater than either individual mutation. As before, WT polymerase activity in DKO cells was not rescued by overexpression of

huANP32E, and we postulate this is at least partly explained by sequence variation at residues 125 and 129 in LRR5 (Leucine Rich Repeat 5), a domain of ANP32 we and others have previously shown important for the polymerase-ANP32 interaction (Supplemental Fig. 3)[9,11].

## ANP32E is essential for polymerase activity and viral replication of the PB1 K577E and PA Q556R mutant

To demonstrate that adaptation to use huANP32E was sufficient to explain replication of the mutant virus in DKO cells, we used CRISPR/Cas9 genome editing to generate eHAP cells lacking functional huANP32A, huANP32B and huANP32E (TKO cells). We designed gRNAs to create adjacent double strand breaks in DKO cells to disrupt the ANP32E locus (Fig. 3A). We confirmed that the TKO clone lacked an intact open reading frame for ANP32E expression through Sanger sequencing revealing the expected deletion. This resulted in a prematurely terminated ANP32E with the first 54 residues as in the WT protein followed by a frameshift mutation caused by the deletion resulting in 28 nonsense residues followed by 2 stop codons (Fig. 3A). A control clone was also chosen that underwent CRISPR/Cas9 editing but still had full length ANP32E.

We tested whether WT or K577E + Q556R polymerase showed activity in TKO cells. As with DKO cells, there was no activity of WT polymerase in TKO cells and its activity could be rescued by huANP32B but not huANP32E (Fig. 3B). Again higher expression levels of PB1 K577E were detected compared to PB1 WT (Fig. 3B) but the K577E + Q556R polymerase proved inactive in TKO cells. However, activity could be rescued by exogenous expression of either huANP32B or huANP32E. Polymerase complexes reconstituted with the individual K577E or Q556R mutations both showed no activity in TKO cells but could be partially rescued by huANP32E.

Next, we tested whether any viral replication was possible in TKO cells. We measured replication of Tky05 and the PB1 K577E + PA Q556R double mutant virus over 72 h in DKO and TKO cells. As before, the double mutant replicated well on DKO control cells, and we detected low but increasing titres in these cells following infection with the WT virus at 48 and 72 h suggesting that further escape mutants were evolving. In contrast, there was no virus detected after infection of TKO cells for either WT or for the double mutant. (Fig. 3C).

To test whether exogenous expression of ANP32E could rescue viral replication in TKO cells, we transfected TKO cells with plasmids for huANP32B, huANP32E or chANP32B and infected with a high MOI of virus (Fig. 3D). Neither WT nor mutant virus gave any virus yield at 72 h post infection of the TKO cells in the absence of exogeneous ANP32. Addition of huANP32B supported viral replication to levels around $10^4$ PFU/ml for both viruses. Moreover, addition of exogenous huANP32E rescued viral growth for K577E + Q556R virus and high titres were produced. In the presence of huANP32E, the WT virus also gave rise to low levels of virus at this late time point. The addition of chANP32B also supported replication of the mutant virus but not WT. Thus, our results demonstrate that the combination of mutations at PB1 K577E and PA Q556R resulted in a virus that was able to replicate through evolving to use suboptimal ANP32 proteins such as huANP32E.

## Q556R is located near PB2 627 in the asymmetric influenza polymerase dimer

Available structures of the influenza polymerase heterotrimer were used to examine the locations of PB1 577 and PA 556. PB1 residue 577 is located in the thumb domain of the polymerase and close to the mode B site in PB1 responsible for RNA binding[34]. PA residue 556 is located near one of the binding pockets for the C-terminal domain of cellular RNA PolII[35]. Recently, human and chicken ANP32 have been shown to interact with an asymmetric dimer of influenza polymerase[15]. In the asymmetric dimer, the RNA exit channel of the replicating polymerase

is aligned directly with the entry channel of the 'encapsidating' polymerase[15]. These two polymerases are bridged by the LRR of ANP32 with the LCAR tail positioned between the two PB2 627-domains. PA residue 556 in the encapsidating polymerase is located near to both PB2 627 residues and to ANP32, making it possible that the switch in PA to 556R could provide additional positive charge (similarly to E627K) to further stabilize the interaction between polymerase and the LCAR tail of the host factor (Fig. 4A). Residue 577 in PB1 is not located close to ANP32 in the asymmetric dimer.

## PB1-K577E is important for binding and utilization of ANP32E

To determine whether the ability of the K577E + Q556R polymerase to co-opt ANP32E is due to an enhanced ability to bind to this alternative host factor, we conducted co-precipitation experiments. DKO cells were transfected with plasmids encoding either the WT or K577E + Q556R polymerase and with either FLAG-tagged huANP32B, chANP32B, huANP32E or GFP-FLAG used as a negative control. After cell lysis, proteins were precipitated using anti-FLAG affinity gel and the amount of PB2 was analysed by immunoblot (Fig. 4B). Under these conditions there was no difference in the amount of PB2 that co-precipitated with any of the ANP32 proteins above the GFP control in the context of WT polymerase. However, the K577E + Q556R polymerase did co-precipitate above background levels (GFP control) with huANP32B and huANP32E, but not chANP32B, which matched the ability of the two huANP32 proteins to rescue the mutant polymerase activity in DKO cells (Fig. 2C).

To measure quantitatively the interaction between influenza polymerase and ANP32 proteins, we conducted a split luciferase experiment whereby the N-terminal fragment of Gaussia luciferase (Gluc1) is fused via a linker to C-terminus of the PB1 subunit and the C-terminal fragment (Gluc2) is fused via a linker to the C-terminus of ANP32[11] (Fig. 4C). This interaction assay most likely measures the interaction between ANP32 and the encapsidating influenza polymerase since we previously showed that mutations in ANP32 at amino acids 129 and 130, that map to the encapsidating polymerase: host factor interface in the structure, abrogate the luciferase signal[11]. We compared the signal generated by ANP32 with the WT and either the double or single mutant polymerases (Fig. 4D). For the WT polymerase, we observed a luciferase signal above background only with huANP32B despite similar levels of expression of all ANP32-Gluc2 homologs. The interaction between the double mutant K577E + Q556R polymerase and huANP32B was significantly higher than WT and this was also true for both of the single mutants K577E and Q556R, without changes in protein levels of PA or PB1-Gluc1 proteins. In addition, both double and single mutants interacted with huANP32E, although the signal was lower than for huANP32B interaction, and the binding appeared to be largely driven by the PB1 mutation. Overall, we conclude that the combination of PB1 K577E + PA Q556R allows stronger interaction with huANP32B as well as huANP32E and that the increased binding to huANP32E is mostly driven by K577E. These results are in line with the polymerase activity measured by minigenome assays where PB1 577E polymerase was more readily complemented by huANP32E (Fig. 2C). Given that PB1 residue 577 is located distally from ANP32 in the structure of the asymmetric dimer, K577E must be enabling use of ANP32E by an alternative mechanism.

## K577E decreases dimerization of the polymerase

Next, we mapped the location of PB1 K577E and PA Q556R on the symmetric polymerase dimer[24]. Both mutations reside directly in the dimer interface with the side chains of K577 and Q556 14–16 Å apart (Fig. 5A). Given their location, we investigated whether these mutations alone or in combination altered dimerization of the polymerase. We performed a split luciferase complementation assay in which either the N-terminal or C-terminal fragment of Gaussia luciferase was fused to the C-terminus of a PB1 subunit (Fig. 5B). Tagging the PB1 protein at

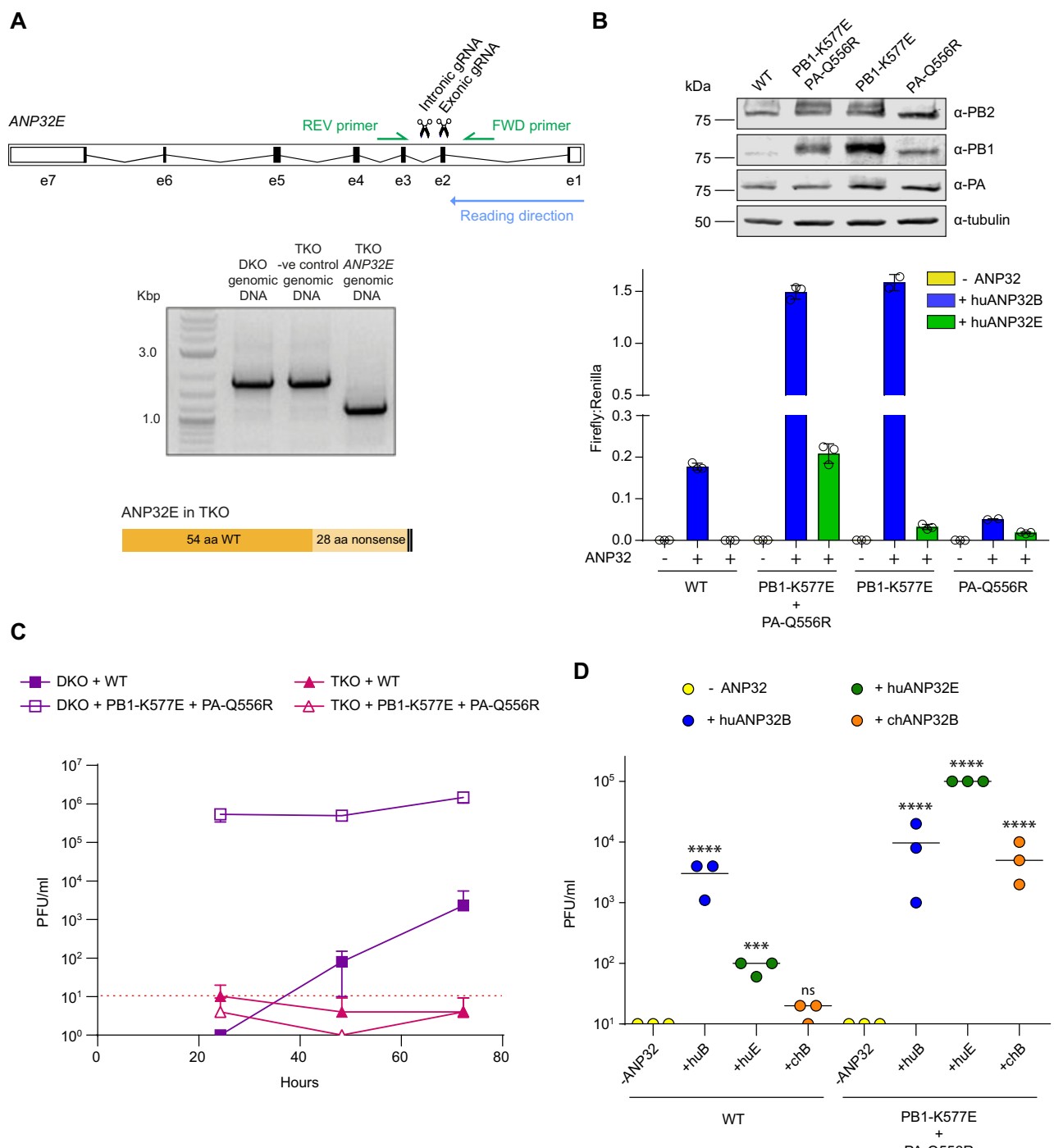

**Fig. 3 | PB1 K577E and PA Q556R allow appropriation of human ANP32E.**
**A** Schematic showing the location of the CRISPR guide RNAs and PCR primers in *huANP32E*. Gel showing PCR product from genomic DNA for WT, −ve control and TKO cells and a schematic illustrating the resulting truncated ANP32E in TKO cells. A single TKO clone was obtained. **B** Minigenome assay conducted in 24-well plates of eHAP TKO cells transfected with pCAGGS Tky05 PB1 or K577E (0.04 µg), PB2 (0.04 µg), PA or Q556R (0.02 µg), NP (0.08 µg), reporter pPolI-luc (0.08 µg) and control pCAGGS-*Renilla* luciferase (0.04 µg) and +/− huANP32B-FLAG (0.04 µg). Data presented is representative of $n = 3$ biological repeats each conducted with $n = 3$ wells, presented as mean values +/− SD. Western blot showing expression of PB2, PB1, PA and tubulin. **C** Viral Growth curve in DKO and TKO cells for WT and PB1 K577E + PA Q556R virus. Six-well plates were infected with virus at MOI of 0.01 and samples plaqued at 24, 48 and 72 h, $n = 3$ wells presented as mean values +/− SD,

limit of detection = 10 PFU/ml. **D** TKO cells were transfected in a 12-well plate with -ANP32, huANP32B, huANP32E or chANP32B (0.32 µg). After 24 h, cells were infected with either WT (Tky05) or PB1 K577E + PA Q556R virus at MOI of 1. Virus was titred at 72 h, $n = 3$ wells. Statistical significance was determined by one-way ANOVA of log transformed data. Samples were compared to -ANP32 of the appropriate virus, $*P < 0.05$, $**P < 0.01$, $***P < 0.001$, $****P < 0.0001$ 3D; Tky WT -ANP32 vs Tky WT +huANP32B $p < 0.0001$, Tky WT -ANP32 vs Tky WT +huANP32E $p = 0.0005$, Tky WT -ANP32 vs Tky WT +chANP32B $p = 0.2189$, Tky PB1-K577E + PA-Q556R -ANP32 vs Tky PB1-K577E + PA-Q556R +huANP32B $p < 0.0001$, Tky PB1-K577E + PA-Q556R -ANP32 vs Tky PB1-K577E + PA-Q556R +huANP32E $p < 0.0001$, Tky PB1-K577E + PA-Q556R -ANP32 vs Tky PB1-K577E + PA-Q556R +chANP32B $p < 0.0001$. Source data are provided as a Source Data file.

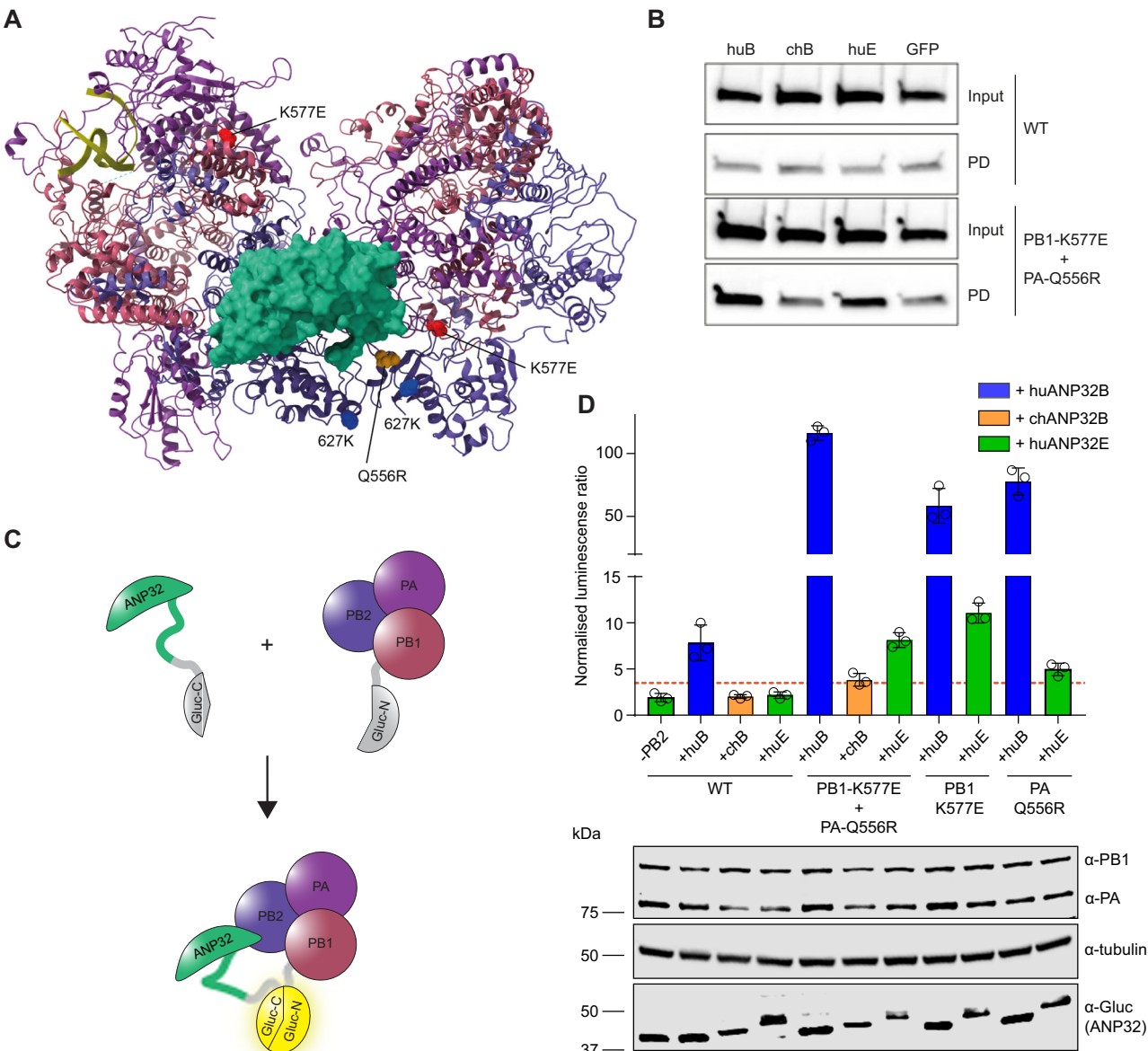

**Fig. 4 | PB1 K577E and PA Q556R increase binding to huANP32E. A** Asymmetric dimer of influenza polymerase (PDB: 6XZP) and ANP32 (green) showing PB2 627K in blue, PB1 K577E in red and PA Q556R in yellow. **B** Co-immunoprecipitation assays were conducted from 10 cm dishes of eHAP TKO cells transfected with pCAGGS Tky05 PB2 (5 μg), Tky05 PB1 or K577E (5 μg), Tky05 PA or Q556R (5 μg) and huANP32B-FLAG/chANP32B-FLAG/huANP32E-FLAG (5 μg). Twenty-four hours after transfection, cells were lysed, FLAG-tagged proteins were immunoprecipitated, and co-precipitation of PB2 was detected using immunoblotting. Data presented are representative of *n* = 3 biological repeats. **C** Schematic showing the reconstitution

of *Gaussia* luciferase activity from the interaction of the C-terminus of *Gaussia* luciferase fused to ANP32 with the N-terminus of *Gaussia* luciferase fused to PB1. **D** Split luciferase assay measuring the interaction between influenza polymerase combinations Tky05 WT, PB1 K577E + PA Q556R, PB1-K577E and PA-Q556R (formed using PB1-Gluc1 or PB1-K577E-Gluc1 fusions) and either huANP32B-Gluc2, chANP32B-Gluc2 or huANP32E-Gluc2. Data presented are representative of *n* = 3 biological repeats each conducted with *n* = 3 wells, presented as mean values +/− SD. Accompanying Western blot showing expression of PA/Q556R, PB1/K577E-Gluc1, tubulin and ANP32-Gluc2. Source data are provided as a Source Data file.

the C-terminus with the Gluc1 or 2 fragments did hinder the activity of the polymerase but the effect was comparable between PB1-WT and PB1-K577E (Supplemental Fig. 4). When the polymerase forms a dimer, the two differently tagged PB1 proteins from each protomer will be brought into close proximity allowing reconstitution of a functional *Gaussia* luciferase. We reasoned that this PB1-PB1 split luciferase complementation assay measured the symmetric but not the asymmetric dimer since in the symmetric dimer, the C-terminals of PB1 are on the same face of the dimer 110 Å apart allowing for the two luciferase fragments to interact, whereas, in the asymmetric dimer, although the C-terminals of PB1 are not both fully resolved, they are on opposite sides of the dimer (~180 Å apart) making the reconstitution of

active luciferase from this pairing unlikely. To experimentally test this reasoning, we introduced alanine mutations in PA 352-6, which have previously been shown to disrupt the symmetric dimer interface[24]. Indeed, we did not detect any significant interaction between PB1 components of a polymerase complex reconstituted with this PA mutant above background levels (in the absence of PA) (Fig. 5C). In contrast, dimerization of WT polymerase was indicated by a strong luciferase signal. That we detected this interaction in TKO cells indicates that the dimerization measured was not dependent on ANP32 proteins. However, when the polymerase contained the PB1 K577E mutations in both tagged PB1 clones either alone or in combination with PA Q556R, there was no signal above background. The PA mutant

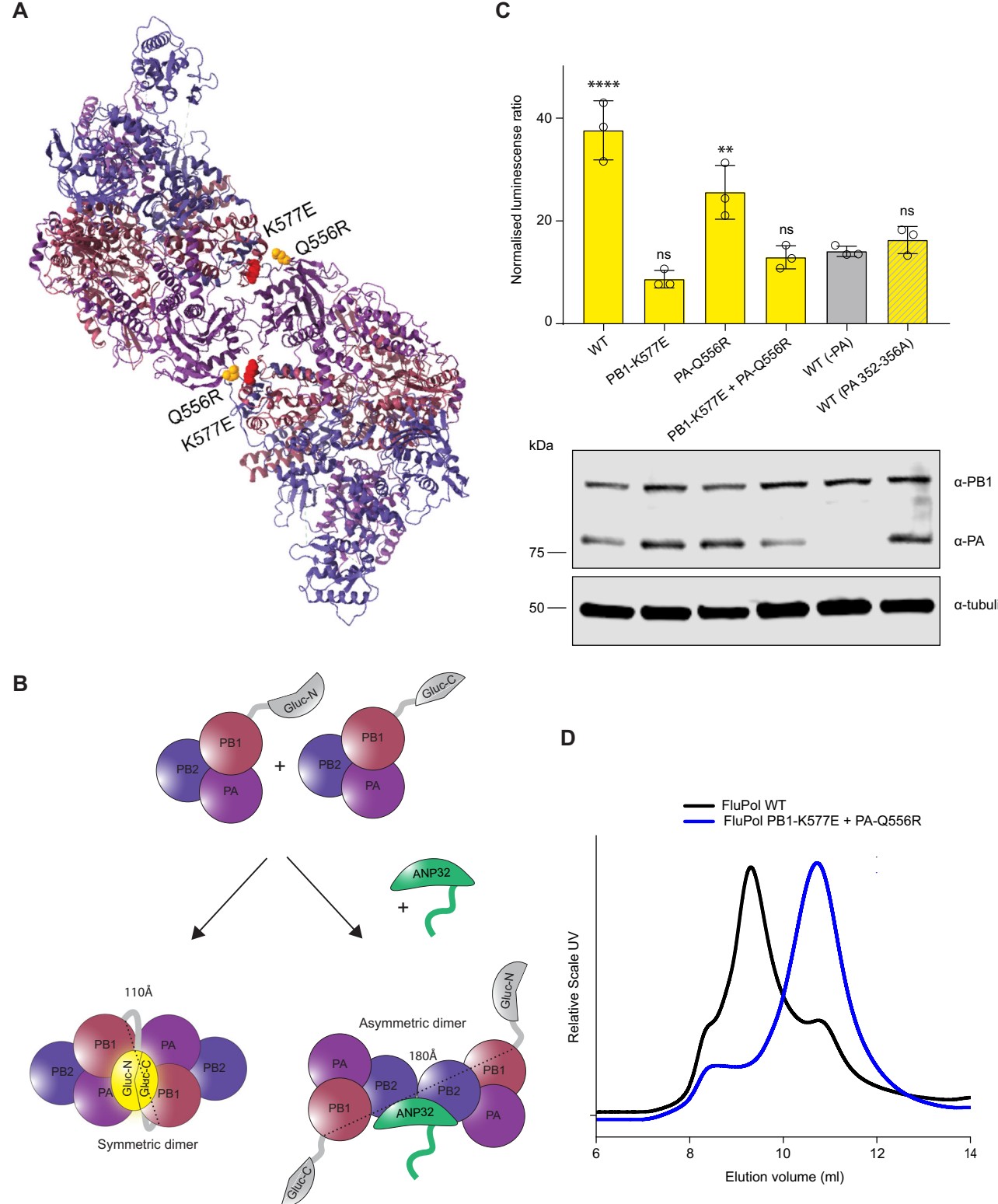

alone had a smaller effect on disrupting the dimerization compared to PB1 K577E.

To further validate how these mutations impact polymerase oligomerization, we purified recombinant WT and mutant polymerases following their expression in insect cells and analysed their elution profiles by size exclusion chromatography (Fig. 5D). WT polymerase eluted with a higher molecular weight peak corresponding to dimers, whereas K557E + Q5565 polymerase eluted later, indicative of monomeric polymerase. Together these results demonstrate that K577E, and

to a lesser extent Q556R, abrogate the formation of the symmetric dimer.

## ANP32E is required for vRNA synthesis by K577E + Q556R virus in the absence of ANP32A and B

Next, we tested how the WT and K577E + Q556R viruses differ in the production of different RNA species in the presence or absence of ANP32 proteins. We measured mRNA, cRNA and vRNA levels in eHAP, DKO and TKO cells for WT and mutant virus at early (2 h.p.i.) and later

**Fig. 5 | PB1 K577E destabilises formation of the polymerase dimer. A** Symmetric dimer of influenza polymerase (PDB: 6QX8) showing PB1 K577E in red and PA Q556R in yellow. **B** Schematic showing the split-luciferase dimerization assay with C-terminus of *Gaussia* luciferase and the N-terminus of *Gaussia* luciferase attached to separate PB1 proteins. The two halves of the luciferase are 110Å in the same plane of the symmetric dimer and 180 Å on opposite sides of the asymmetric dimer. **C** Split luciferase complementation assay measuring dimerization of Tky05 WT, PB1 K577E + PA Q556R, PB1 K577E and PA Q556R polymerases (formed using equal amounts of PB1-Gluc1 + PB1-Gluc2 or PB1-K577E-Gluc1 + PB1-K577E-Gluc2). Data presented are representative of $n = 3$ biological repeats each conducted with $n = 3$ wells, presented as mean values +/− SD. Statistical significance was determined by multiple comparisons of one-way ANOVA, samples were compared to WT (-PA), *$P < 0.05$, **$P < 0.01$, ***$P < 0.001$, ****$P < 0.0001$ (-PA vs WT $p < 0.0001$, -PA vs PB1-K577E $p = 0.2708$, -PA vs PA-Q556R $p = 0.0019$, -PA vs PB1-K577E + PA-Q556R $p = 0.9993$, -PA vs WT(PA 352-356A) $p = 0.9683$). Accompanying Western blot showing expression of PB1/K577E-Gluc, PA/Q556R and tubulin. **D** Tky05 WT polymerase and PB1 K577E + PA Q556R were cloned into baculovirus and expressed in Sf9 cells. The polymerases were purified and concentrated (as described in the methods) then analysed by size exclusion chromatography. Lower elution volume is indicative of higher molecular weight. Source data are provided as a Source Data file.

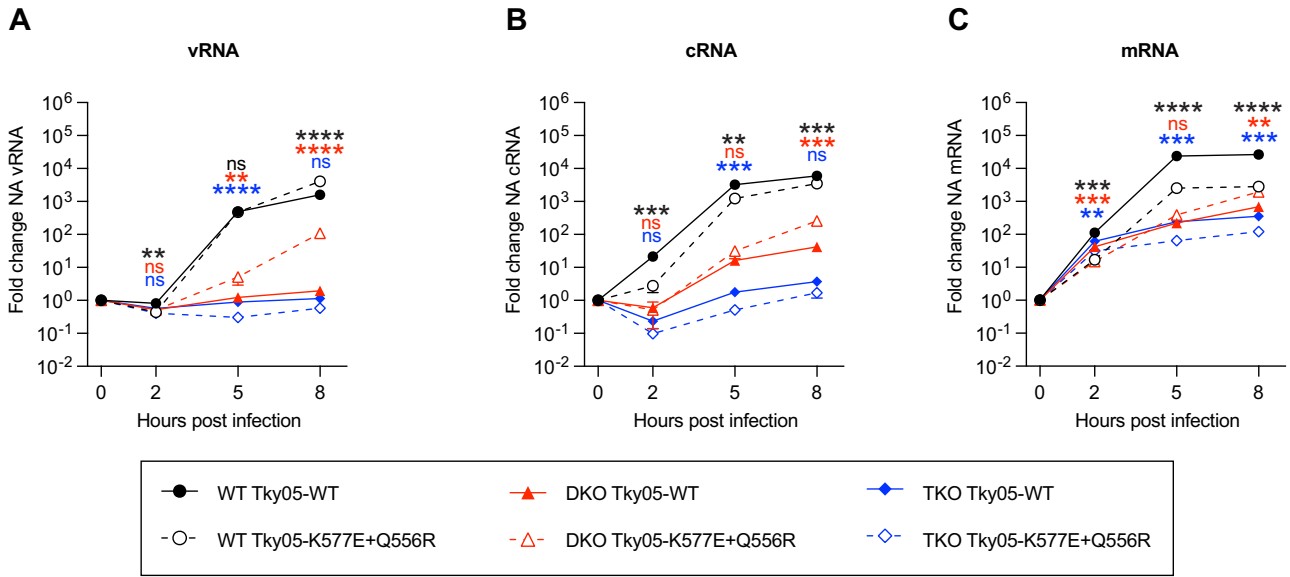

**Fig. 6 | PB1 K577E + PA Q556R requires ANP32E to support vRNA synthesis. A–C** Segment 6 vRNA (**A**) cRNA (**B**) or mRNA (**C**) accumulation over time in eHAP WT, DKO and TKO cells following infection with Tky05-WT or Tky05-K577E + Q556R (MOI = 3). Fold change was calculated over input (0 h.p.i.) for each cell type. $n = 3$ biological replicates, plotted as mean ± s.d. Statistical significance was assessed using multiple unpaired two-sided $t$ tests following log transformation, corrected for multiple comparisons using the false discovery rate *$P < 0.05$, **$p < 0.01$; ***$p < 0.001$; ****$p < 0.0001$. Statistical comparisons are between Tky05-WT and Tky05-K577E + Q556R within the same cell line at each timepoint (vRNA WT cells; 2 h, $p = 0.0043$, 5 h $p = 0.7484$, 8 h $p < 0.0001$, vRNA DKO cells; 2 h $P = 0.8564$, 5 h $p = 0.0043$, 8 h $p < 0.0001$, vRNA TKO cells; 2 h $p = 0.0464$, 5 h $p < 0.0001$, 8 h $p = 0.0266$, cRNA WT cells; 2 h $p = 0.009$, 5 h $p = 0.0015$, 8 h $p = 0.0004$, cRNA DKO cells; 2 h $p = 0.4967$, 5 h $p = 0.0539$, 8 h $p = 0.0102$, cRNA TKO cells; 2 h $p = 0.0444$, 5 h $p = 0.0004$, 8 h $p = 0.0102$, mRNA WT cells; 2 hrs $p = 0.0003$, 5 h $p < 0.0001$, 8 h $p < 0.0001$, mRNA DKO cells; 2 h $p = 0.0003$, 5 h $p = 0.0259$, 8 h $p = 0.0010$, mRNA TKO cells; 2 h $p = 0.0035$, 5 h $p = 0.0004$, 8 h $p = 0.0002$). Comparisons for WT cells are shown at the top, DKO cells in the middle and TKO cells at the bottom (indicated by matching colour to traces). Source data are provided as a Source Data file.

(5 and 8 h.p.i.) time points (Fig. 6). At 2 h.p.i., mRNA resulting from primary transcription was detected in all cells and for both viruses, in line with previous observation that ANP32 proteins do not impact transcription[8,36]. vRNA replication for both viruses was evident by 5 h.p.i., and levels increased further by 8 h.p.i. In contrast, in DKO or TKO cells, vRNA levels of WT virus did not increase whereas the mutant virus showed a significant increase in vRNA by 8 h.p.i. in DKO cells, but not in TKO cells. This confirms that ANP32 proteins are required for vRNA synthesis and that the mutant virus can use ANP32E in DKO cells to accomplish this step of replication. Interestingly, we observed some increase in cRNA levels in DKO cells even for WT virus in contrast to our recently reported results[36]. This may underlie the ability of this particular strain to evolve in the way we have described here.

### Murine ANP32 proteins are suboptimal for support of influenza polymerase but K577E and Q556R increase polymerase activity and virulence in mice

A literature search revealed that K577E[37] and Q556R[38–43] have both previously been recognized separately as mammalian adaptations of avian influenza viruses in mice and have also occurred together in a mouse-adapted H3N2 virus[39]. This suggested that K577E and Q556R could also serve as adaptations for influenza A virus polymerase to use murine ANP32 proteins. Previously, only mouse ANP32B has been shown to be able to support influenza A virus replication[8,9,44]. We performed a minigenome assay in DKO cells comparing activity of WT to K577E + Q556R polymerase in the presence of exogenously expressed muANP32A, muANP32B or muANP32E (Fig. 7A). All ANP32 proteins were well expressed and exogenous ANP32 expression did not affect levels of polymerase proteins, indicated by immunoblot of PB2. For WT polymerase, murine ANP32 proteins provided little support of activity, with the muANP32B being the only one to result in luciferase reporter signals above background. This is in keeping with murine ANP32 proteins harbouring sequence differences in LRR5 especially amino acids 129 and 130 previously mapped to interact with ANP32 (Supplemental Fig. 3). However, the K577E + Q556R polymerase was active in the DKO cells, and activity was further enhanced by exogenous expression of either muANP32A, muANP32B or muANP32E, with muANP32B being most potent (Fig. 7A). This effect was driven mostly by the K577E mutation in line with the earlier results with human ANP32 proteins (Fig. 2).

To test whether the K577E/Q556R virus replicated better in mice, we infected mice with WT Tky05 or K577E/Q556R virus. Mice infected

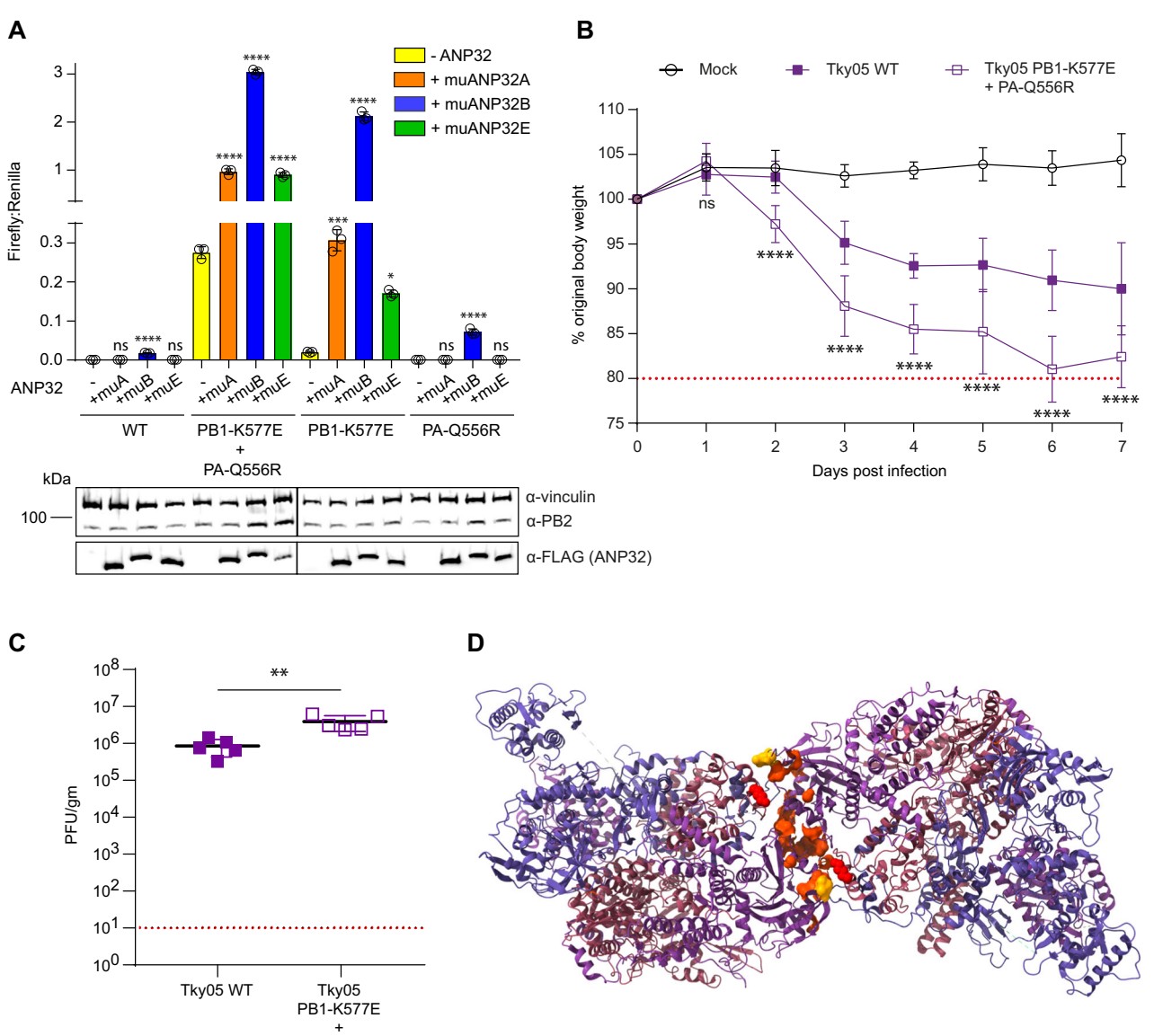

**Fig. 7 | PB1 K577E + PA Q556R polymerase can use suboptimal murine ANP32 proteins. A** Minigenome assays were performed in 24-well plates of eHAP DKO cells transfected with pCAGGS Tky05 PB2 (0.04 μg), PB1 or K577E (0.04 μg), PA or Q556R (0.02 μg), NP (0.08 μg), reporter pPolI-luc (0.08 μg), control pCAGGS-*Renilla* luciferase (0.04 μg) and either -ANP32, muANP32A-FLAG, muANP32B-FLAG or muANP32E-FLAG (0.04 μg). Data presented are representative of *n* = 3 biological repeats each conducted with *n* = 3 wells, presented as mean values +/− SD. Statistical significance was determined by multiple comparisons of one-way ANOVA, *P* < 0.05, **P* < 0.01, ***P* < 0.001, ****P* < 0.0001 (Tky WT; -ANP32 vs +muANP32A p = 0.923, -ANP32 vs +muANP32B p > 0.0001, -ANP32 vs +muANP32E *p* = 0.923, Tky PB1-K577E + PA-Q556R; -ANP32 vs +muANP32A *p* < 0.0001, -ANP32 vs +muANP32B *p* < 0.0001, -ANP32 vs +muANP32E *p* < 0.0001, Tky PB1-K577E; -ANP32 vs +muANP32A p = 0.0003, -ANP32 vs +muANP32B *p* < 0.0001, -ANP32 vs +muANP32E *p* = 0.0135, Tky PA-Q556R; -ANP32 vs +muANP32A *p* < 0.9999, -ANP32 vs +muANP32B *p* < 0.0001, -ANP32 vs +muANP32E *p* > 0.9999). Accompanying western blot showing expression of vinculin, PB2 and ANP32-FLAG. **B** Six to eight-week-old female BALB/c mice were mock infected (10 mice) or infected intranasally with $10^5$ PFU of Tky05 WT (10 mice) or PB1 K577E + PA Q556R virus (10 mice). 5 mice per group were culled at day 2 and 5 mice culled at day 7. Weight loss was measured each day. One mouse was excluded from the Tky05 WT group for weight loss calculation because it did not become infected after inoculation. Data are presented as mean values +/− SD. Statistical significance was determined by two-way ANOVA, *P* < 0.05, **P* < 0.01, ***P* < 0.001, ****P* < 0.0001 (displayed is a comparison between WT and PB1-K577E + PA-Q556R; day 1 p > 0.3258, day 2 p < 0.0001, day 3 p < 0.0001, day 4 p < 0.0001, day 5 p < 0.0001, day 6 p < 0.0001, day 7 p < 0.0001). **C** Viral titres from the homogenized lung tissue on day 2 p.i. (*n* = 5 murine lungs per group). Data are presented as mean values +/− SD. Statistical significance was determined by an unpaired *t*-test, *P* < 0.05, **P* < 0.01, ***P* < 0.001, ****P* < 0.0001 (WT vs PB1-K577E + PA-Q556R p = 0.0058). **D** Symmetric dimer of the influenza A polymerase (PDB: 6QX8) with mouse-adaptive mutations mapping to the interface. Highlighted residues are PA N291, L295, L336, A343, D347, E349, K353, K356, L550, T552, I554, Q556 and PB1 K577. Source data are provided as a Source Data file.

with the K577E/Q556R virus showed significantly greater weight loss compared to Tky05 infection with several K577E/Q556R virus-infected mice reaching the humane endpoint before 7 days (Fig. 7B). In agreement with weight loss data, K577E/Q556R virus titre was significantly higher than for Tky05 in the lungs at day 2 post infection (Fig. 7C).

To investigate whether other mutations found in avian influenza viruses that have adapted to mice could also be increasing the use of

mouse ANP32 proteins through a similar mechanism that affects polymerase dimerization, we mapped murine adaptations described in the literature onto the structure of the symmetric dimer (Fig. 7D). We found many examples of mouse-adapting mutations in the symmetric dimer interface, most notably mutations at PA N291[45], L295[46], L336[47], A343[48–52], D347[51,53,54], E349[43,45,55–57] (already shown to be a mutant that reduces polymerase dimerization[28]), K353[58], K356[52,59] and other sites

referenced in[24]. In addition, there were several reported mutations close to PA Q556 at positions 549–554, which could affect either polymerase dimerization or mediate a more direct interaction with ANP32[43,52,60–63].

## Discussion

ANP32 proteins are crucial host factors for influenza replication. In human cells, huANP32A or huANP32B are required to support replication of influenza A viruses, whereas huANP32E is usually inactive. After passage on human cells lacking huANP32A and huANP32B, an influenza A virus evolved to use huANP32E. The evolved virus did not replicate in the complete absence of ANP32 proteins, which confirms their essential role in influenza propagation. huANP32E and chANP32B were previously found incapable of supporting influenza A polymerase, although overexpressed huANP32E did exhibit limited support for influenza B polymerase activity[64]. Both these ANP32 proteins differ from those that do support the viral polymerase at amino acids 129 and/or 130. The I129/N130 pair were mapped as key residues preventing chANP32B from supporting influenza polymerase[9,11]. huANP32E has E129 instead of the N129 found in the huANP32A and B proteins, which is likely responsible for its inability to support WT influenza polymerase, and murine ANP32 proteins also vary at these residues accounting for their suboptimal use by the WT virus as well (Figs. 2D and 7B).

The recombinant virus with internal genes from Tky05 was chosen as the ancestor virus for these experiments due to low levels of replication observed on DKO cells. It is currently unknown whether other viruses would evolve to use ANP32E as readily. Interestingly, although we recently showed that for most strains, ANP32 proteins are required to support stabilization of cRNA produced in the pioneering round of replication from incoming vRNA genomes[36], here we observed some increase in cRNA levels in DKO cells even for the ancestral virus. This may underlie the ability of this strain to evolve to use ANP32E. Another recent observation suggested that NEP proteins of influenza A can compensate for suboptimal ANP32 use[65], which suggests that variation in NEP could help determine which viruses can adapt to suboptimal ANP32s.

We focused here on an evolved viral clone with two polymerase mutations, PB1 K577E and PA Q556R. K577E and Q556R were found together on a clone from a single population but were not present in the consensus sequence of two other Tky05 virus escape populations, which also had increased growth on DKO cells (Fig. 1B, C). This suggests there are likely other evolutionary paths towards utilization of huANP32E and further studies will aim to elucidate these. The K577E/Q556R combination allowed polymerase to interact with and use huANP32E, but not chANP32B, at least in the polymerase assay

(Fig. 2C). huANP32E, and all 3 murine ANP32 proteins have a single mutation at 129/130 compared to huANP32A/B whereas chANP32B is different at both sites (Supplemental Fig. 3). The additional change may make it harder for the influenza polymerase to appropriate chANP32B for replication. Other changes elsewhere in chANP32B may also reduce its ability to interact with influenza polymerase and support activity[11].

PB1 K577E and PA Q556R reside in the symmetric dimer interface that is formed between two polymerase heterotrimers. This dimer is implicated in the replication step of cRNA to vRNA synthesis, a step for which ANP32 has been shown to be required. Indeed, our results here confirm the essential role of ANP32 proteins for vRNA production (Fig. 6A). A previous study identified PB1 K577G as a mutation which reduced influenza A polymerase dimerization and restored fitness to a reassortant virus that had a polymerase with a mixed origin of subunits[28]. A more recent study found that the adjacent residue K578 is subject to ubiquitinylation and that mutation to A or R at that site impacts dimerization, polymerase activity and replication[27]. Here we found that K577E on its own reduced dimerization of the Tky05 polymerase. However, K577E alone did not allow polymerase activity in the DKO cells implying that reduced dimerization of the symmetric dimer by itself does not efficiently support the use of ANP32E without further changes. Although Q556R is present in the symmetric dimer interface, it did not impair dimerization to the same extent as PB1 K577E. However, like PB1 K577E, PA Q556R did enhance the polymerase's interaction with both ANP32 B and E. Considering the proximity of PA Q556R to the C-terminus of ANP32, it seems possible that this charge reversal mutation makes direct interaction with the acidic residues of the ANP32 LCAR. We propose that a reduction in dimerization is a key component of the ability of the influenza virus to use an alternative ANP32, but the polymerase may require additional mutations to use ANP32E with robust activity.

In the only structure of polymerase in complex with ANP32, the host protein binds across the replicating and encapsidating polymerase heterotrimers that form an asymmetric dimer. ANP32 residues 129 and 130 form contacts with the polymerase in this structure, but PB1 577 and PA 556 are not located close to these contacts. PA 556 is proximal to PB2 627 and, as mentioned above, could interact with the LCAR of ANP32. However, this still raises the question of why we measure increased interaction between the PB1 K577E/PA Q556R mutated polymerase and ANP32. Our observations could be explained by the escape mutations altering a balance between the two alternate forms of polymerase dimers (Fig. 8). By reducing the ability of the polymerase to form the symmetric dimer, the virus favours the formation of the asymmetric dimer even with a suboptimal ANP32. Thus the balance between monomeric (transcriptionally competent),

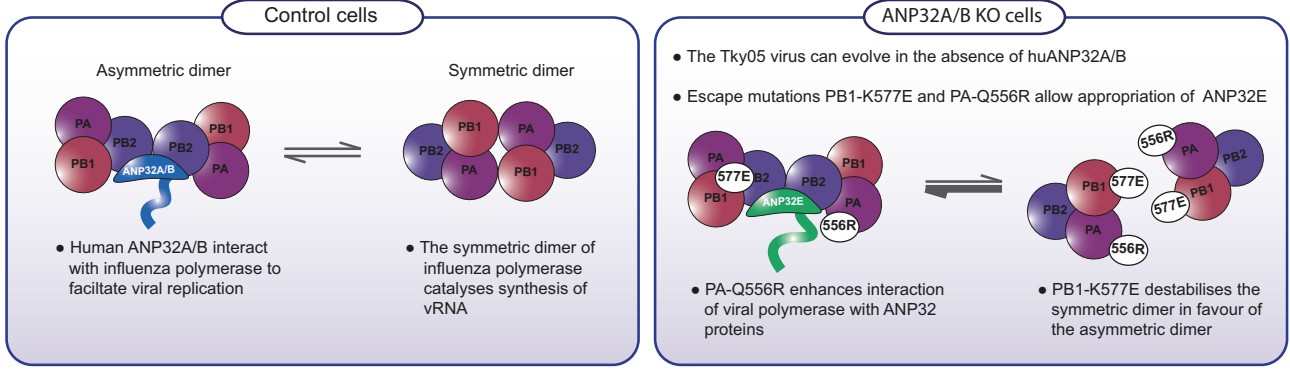

**Fig. 8 | PB1 K577E and PA Q556R rebalance influenza polymerase dimer formation.** Schematic showing the proposed mechanism of how PB1 K577E and PA Q556R appropriate ANP32E to re-balance the equilibrium of polymerase dimers.

asymmetric and symmetric dimers is dictated by the relative stoichiometry of FluPol and ANP32 proteins. One caveat to this study is that our interaction assays rely on overexpressed levels of protein that may not reflect the levels of endogenous ANP32 or changing abundance of FluPol throughout infection. More research will be needed to fully understand the balance of polymerase dimers and how this impacts on polymerase mode of action.

E627K in PB2 is a common mammalian adaptation in avian influenza viruses that arises to complement the mammalian version of ANP32[10]. Further research has linked other mutations in PB2 to ANP32 adaptation[19] and has also suggested that certain mutations in PA could facilitate the acquisition of the E627K mutation[66]. Our results now show that mutations in PB1 and PA can also be involved in adaptation to enhance utilization of otherwise suboptimal ANP32 proteins. Both K577E and Q556R have been found to adapt influenza viruses to mice[37–43]. We found that mouse ANP32 proteins were surprisingly poor at supporting influenza polymerase in a minigenome assay likely due to the amino acid sequences of murine ANP32A, ANP32B and ANP32E at position 129/130. Both WT and mutant polymerases were most efficiently complemented by murine ANP32B which agrees with results showing that ANP32B is required for viral pathogenicity in mice[44]. K577E increased polymerase activity in conjunction with all murine ANP32 proteins, and the K577E/Q556R virus showed increased replication and pathogenicity in mice. Q556R alone also showed an increase in polymerase activity with muANP32B. Supporting our hypothesis that reduced polymerase dimerization promotes use of suboptimal ANP32 proteins, many previously described mouse-adapting mutations map to the symmetric dimer interface (Fig. 7E)[24]. One of these mutations, E349K, has already been shown to reduce formation of the symmetric dimer[28] and we predict other mutations will show a similar effect.

Host factors that are co-opted by viruses to support their replication are promising targets for new drugs and for gene editing to create virus resilient farmed animals such as poultry or swine. Our results demonstrate the importance of performing experimental evolution to test the robustness of such blocks as the virus may be capable of adaptation to use an alternative host factor. A previous study examining whether influenza could evolve to replicate in mice lacking importin-α7 also described rapid viral adaptation and found that viruses had also become more virulent in WT mice[30]. It is not surprising that a virus as successful as influenza at infecting different hosts has the flexibility to adapt to use alternative host factors. It will be important to discover all the alternative pathways the virus can use, and whether adaptation comes with a fitness cost, in order to establish whether host targeting strategies are a viable path to virus control. The K577E/Q556R mutations we studied here had no obvious fitness cost in eHAP or in MDCK cells. The ability of the polymerase to use a suboptimal ANP32 could lead to a trade-off in other aspects of polymerase activity, for example the K577E/Q556R virus showed lower levels of transcription leading to reduced mRNA levels and also lower cRNA compared to WT (Fig. 6). The increased replication of the mutant virus in mice reflects the adaptation for use of murine ANP32 proteins which are otherwise sub-optimally co-opted by the WT Tky05 virus, and does not necessarily equate to an increase in fitness in more relevant hosts. The exact role ANP32 plays in influenza replication is still unclear and the full set of polymerase complexes that form during transcription and replication has yet to be fully described. Experimental evolution, such as described here, is an approach that can contribute to this understanding.

## Methods
### Ethics statement
All work was approved by the local genetic manipulation (GM) safety committee of Imperial College London, St Mary's Campus (centre number GM77), and the Health and Safety Executive of the United Kingdom and carried out in accordance with the approved guidelines. All animal research described in this study was approved and carried out under a United Kingdom Home Office License, P48DAD9B4 in accordance with the approved guidelines.

### Statistics and reproducibility
No statistical method was used to predetermine sample size for in vitro experiments. For animal experiments a resource equation was used to predetermine group size for weight loss measurement. A power calculation, using previous data from mouse infection experiments, was used in G Power to determine group size required to measure differences in virus lung titres with a power of 0.95 or greater. The experiments were not randomized. The Investigators were not blinded to allocation during experiments and outcome assessment. One mouse was excluded from the Tky05 WT infected group for weight loss determination because it did not become infected after inoculation.

### Cells and Virus
Madin-Darby canine kidney (MDCK; ATCC) and HEK293T (293T) were grown in Dulbecco's modified Eagle's medium (DMEM; Invitrogen) supplemented with 10% fetal bovine serum (FBS; labtech.com), 1% penicillin-streptomycin (Invitrogen) and 1% non-essential amino acids (NEAA; Gibco). Human eHAP cells (eHAP; Horizon Discovery) were grown in Iscove's modified Dulbecco's medium (IMDM; Thermo Fisher) supplemented with 10% FBS, 1% penicillin-streptomycin and 1% NEAA. All cells were grown at 37 °C and 5% $CO_2$.

The virus used in this study (Tky05) was a 6:2 reassortant virus with all internal segments from A/turkey/Turkey/1/2005(H5N1) and HA and NA segments from A/Puerto Rico/8/1934(H1N1), rescued from cDNA using reverse genetics techniques as described in Li et al.[33]. This virus was chosen because of preliminary data that indicated a low level of replication in human cells lacking ANP32A and B, and also to mitigate safety concerns because the PR8 derived HA and NA gene segments restrict infection of humans[67] and cross react with antibodies from the pH1N1 component of the seasonal influenza vaccine. Virus stocks were grown on MDCK cells at 37 °C and 5% $CO_2$.

### Viral growth curves
Six- or 12-well plates of confluent cells were infected by viruses at an MOI (multiplicity of infection) between 0.0005 and 0.01 as stated. Cells were maintained in serum-free medium with the addition of 1 µg/ml TPCK trypsin (Worthington-Biochemical). Supernatant samples were taken every 8 h for the first 24 h post infection, and then every 24 h, and frozen until being titred by plaque assay on MDCK cells.

### Experimental evolution
Three wells in a six-well plate of eHAP cells knocked out for ANP32A and ANP32B[8] were infected with 10,000 p.f.u. (plaque forming units) of Tky05 virus. These were the three treatment populations. In addition, three wells of eHAP cells were infected in the same manner as control populations. After three days, the supernatant was centrifuged to remove potential cell debris, frozen and titred. If possible, 10,000 p.f.u. were then used to infect cells for each passage. If the titre was not high enough, then 200 µl of supernatant was used to infect for the next passage. This experiment was performed for four passages.

### Plaque purification
Virus was grown on a six-well plate of MDCKs overlaid with a semi-solid agarose overlay medium[68]. Plaques were picked with a pipette tip before being frozen in 300 µl PBS with 0.35% Bovine Serum Albumin and 1% penicillin-streptomycin. Virus was then grown on six-well plates of MDCKs for initial characterization and further amplified on MDCKs to generate a stock if needed.

## Sequencing

Viral RNA was extracted using QIAmp mini Viral kit (Qiagen). Reverse transcription was performed using SuperScript IV (Invitrogen) and PCR was performed using KOD polymerase (Merck) with gene specific primers for Sanger sequencing (Eurofins).

## Minigenome assay

eHAP DKO or TKO cells were seeded into 24-well plates and transfected with pCAGGS expression plasmids encoding Tky05 PB1 (0.04 µg), PB2 (0.04 µg), PA (0.02 µg), NP (0.08 µg) and ANP32-FLAG (0.04 µg unless otherwise stated) together with the viral reporter plasmid pPolI-luciferase (0.08 µg) and the internal control plasmid pCAGGS-*Renilla* luciferase (0.04 µg). Twenty-four hours after transfection, cells were lysed in 50 µl passive lysis buffer (Promega) for 30 min at room temperature with gentle shaking. Bioluminescence generated by firefly and *Renilla* luciferases was measured using the dual-luciferase system (Promega) on a FLUOstar Omega plate reader (BMG Labtech).

## Immunoblot analysis

Cell pellets were incubated on ice for 1 h in lysis buffer (50 mM Tris-HCl pH 8.0, 150 mM NaCl, 0.5% Na-deoxycholate, 1% Igepal, 0.1% SDS) supplemented with an EDTA-free protease inhibitor cocktail tablet (Roche) and 1 mM PMSF. Cell lysates were centrifuged at $16,000 \times g$ for 15 mins and the supernatant was mixed with Laemmli 4× buffer (Bio-Rad). Protein samples were resolved by SDS-PAGE using Mini Protean TGX precast gels 4–20% (Bio-Rad) and subsequently transferred to nitrocellulose (Amersham Protran Premium 0.2 µm NC; GE Healthcare) or low fluorescence PVDF (Immobilon-FL 0.45 µm; Fisher Scientific) membranes. Membranes were then blocked in 5% milk-TBS for 1 h at room temperature, followed by an overnight incubation at 4 °C with the appropriate primary antibodies: rabbit α-vinculin (Abcam ab129002, 1:2000), mouse α-tubulin (Abcam ab7291, 1:2500), rabbit α-IAV PB2 (GeneTex GTX125926, 1:2000), rabbit α-IAV PB1 (GeneTex GTX125923, 1:500), rabbit α-IAV PA (GeneTex GTX118991 1:500) mouse α-FLAG (Sigma-Aldrich F1804, 1:500), rabbit α-Gaussia luciferase (Invitrogen; PA1-18, 1:1000). Membranes were washed in TBS-1% tween and incubated with secondary antibodies for 1 h at room temperature. Secondary antibodies included sheep anti-rabbit IgG, HRP (Sigma-Aldrich AP510P, 1:20,000) and goat anti-mouse IgG, HRP (Bio-Rad STAR117P 1:1000) or goat anti-rabbit IgG IRdye800CW (Abcam ab216773, 1:20,000) and goat anti-mouse IgG IRdye680RD (Abcam ab216776, 1:20,000). Following washing, protein bands were visualised by chemiluminescence using SuperSignal™ West Femto substrate (Thermofisher Scientific) and/or Odyssey Imaging System (LI-COR Biosciences).

## Generation of triple knockouts/ CRISPR

Protospacer sequences in exon 2 (TAATGTGGAACTAAGTTCGC) and intron 2 (GTACAACTAGAATCCAAGCT) of the ANP32E gene were selected using CCTop[69], and corresponding Alt-R CRISPR-Cas9 RNAs (crRNAs) were obtained from IDT (AltR1/rUrArA rUrGrU rGrGrA rArCrU rArArG rUrUrC rGrCrG rUrUrU rUrArG rArGrC rUrArU rGrCrU /AltR2 and AltR1/rGrUrA rCrArA rCrUrA rGrArA rUrCrC rArArG rCrUrG rUrUrU rUrArG rArGrC rUrArU rGrCrU /AltR2). Cas9-generated DSBs at these sites led to a 615-bp deletion including the splice donor site of exon 2. RNPs containing wild-type Alt-R S.p. Cas9 nuclease V3 (IDT), Alt-R CRISPR-Cas9 tracrRNA (IDT) and crRNAs were reverse transfected into eHAP cells already lacking ANP32A and ANP32B (double knockout)[8], using Lipofectamine CRISPRMAX transfection reagent (Thermo Fisher Scientific). The transfected cells were incubated for 48 h prior to single cell sorting on a FACS Aria III Cell Sorter (BD Biosciences). 7–10 days later, genomic DNA was extracted from single cell-derived clonal populations using a PureLink Genomic DNA Mini Kit (Invitrogen), and the locus of interest was screened by PCR using primers GGATCCGTGTAAGGGGATTGG and GGACATTTCTCTGCCAGG

ACT. Clones with a deletion of the correct size were then verified by Sanger sequencing.

## Complementation of virus with exogenously expressed ANP32

Complementation of virus with exogenously expressed ANP32 was performed by first transfecting TKO cells in a 12-well plate with 320 ng of either huANP32B, huANP32E or chANP32B using Lipofectamine 3000. After 24 h, cells were then infected with either tky05 or K577E/Q556R at an MOI of 1. The inoculum was removed after 1 h and the cells washed twice with PBS adjusted with HCl to lower the pH to 3.0. Further PBS washes were conducted before the addition of DMEM containing 1mg/ml trypsin. Seventy-two hours post infection, viral titre of the supernatant was determined by plaque assay.

## Co-immunoprecipitation

eHAP DKO cells were seeded into 10 cm dishes and transfected with pCAGGS expression plasmids encoding Tky05 PB1 (5 µg), PB2 (5 µg), PA (5 µg) and the indicated ANP32-FLAG (5 µg). Twenty-four hours after transfection, cells were washed in PBS and incubated on a rotating wheel at 4 °C for 30 min in lysis buffer (50 mM Tris-HCl pH 8.0, 150 mM NaCl, and 1% Igepal, supplemented with an EDTA-free protease inhibitor cocktail tablet (Roche). Cell lysates were centrifuged at $16,000 \times g$ for 20 min and the relative total protein content of each supernatant was measured by a nano-spectrophotometer. Following normalisation, the supernatants were incubated with pre-washed anti-flag M2 affinity gel (Sigma-Aldrich) on a rotating wheel at 4 °C overnight. After three washes with TBS at 4 °C, proteins were eluted by the addition of 100 µl of 150 ng/µl 3× FLAG peptide. Co-immunoprecipitated proteins were detected using immunoblot analysis as described above.

## Split Luciferase binding

For measuring ANP32-influenza-polymerase interactions, eHAP DKO cells were seeded into 48-well plates and transfected with pCAGGS expression plasmids encoding Tky05 PB1-Gluc1 (0.04 µg), PA (0.04 µg), PB2 (0.04 µg), and the indicated ANP32-Gluc2 construct (0.04 µg). Control conditions contained pCAGGS-Gluc1 and untagged PB1, or pCAGGS-Gluc2 and untagged ANP32, with all other components remaining constant. Twenty-four hours after transfection, cells were lysed in 60 µl *Renilla* lysis buffer (Promega) for 1 h at room temperature with vigorous shaking. *Gaussia* luciferase activity was assayed using the Renilla luciferase kit (Promega). Injection of substrate and measurement of bioluminescence were carried out using the FLUOstar Omega plate reader (BMG Labtech). Normalized luminescence ratios (NLR) were calculated by dividing the signal from the potential interacting partners by the sum of the two controls[70].

For measuring polymerase dimerization, eHAP TKO cells were transfected with pCAGGS expression plasmids encoding Tky05 PB1-Gluc1 (0.04 µg) or PB1-Gluc2 (0.04 µg) both codon optimised for gene expression in *H. sapien* cells, as well as PA (0.08 µg), PB2 (0.08 µg). The assay was then conducted as detailed above.

## Size exclusion chromatography (SEC)

Polymerase subunits PA, PB1 and protein A-tagged PB2 of A/turkey/Turkey/1/2005 were expressed in Sf9 cells from codon-optimized genes (Synbio) cloned into a single baculovirus using the MultiBac system[71]. Mutations PB1-K577E and PA-Q556R were introduced by mutagenesis PCR. Sf9 cell suspension cultures, maintained in Sf-900 II serum-free medium (Gibco) without antibiotics, were infected with baculoviruses expressing WT or mutant influenza virus polymerase and incubated for 72 h at 27 °C with shaking. Cells were harvested by centrifugation and lysed by sonication (3 × 30 s with 30 s intervals) in buffer A (50 mM Hepes:NaOH (pH 7.5), 500 mM NaCl, 10% (v/v) glycerol, 0.05 (w/v) octylthioglucoside, 1 mM DTT), complemented with protease inhibitors (Roche, cOmplete Mini, EDTA-free) and 100 µg/ml

RNase A. The lysate was clarified by centrifugation (35,000 × *g*, 45 min, 4 °C) and the supernatant was incubated with IgG sepharose beads (Cytiva, IgG Sepharose 6 Fast Flow, 1 ml slurry per L culture) for 3 h at 4 °C. After binding, the beads were washed extensively with buffer A and the polymerase was released overnight at 4 °C with 0.5 mg TEV protease in buffer A. The supernatant containing the polymerase was collected by centrifugation, concentrated to <500 µl and SEC was performed on a Superdex 200 Increase 10/300 GL column (GE Healthcare) in buffer B (25 mM Hepes:NaOH (pH 7.5), 500 mM NaCl, 5% (v/v) glycerol, 1 mM DTT).

### RT-qPCR

For RT-qPCR analysis, eHAP cells were cultured in 24-well plates, with each condition in triplicate. Infections were performed at an MOI of 3. To synchronize infection, viral inoculation was performed at 4 °C. At the appropriate time point, cells were lysed using buffer RLT (Qiagen), frozen at −80 °C, then total RNA extracted using an RNeasy mini kit (Qiagen). Quantification of segment 6 vRNA, cRNA and mRNA was based on a tagged primer approach[72]. For each sample, four reverse transcription reactions were set up using 200 ng RNA/reaction, RevertAid H Minus Reverse Transcriptase, plus a tagged primer targeting either vRNA or cRNA, a tagged polydT (for viral mRNA) or an untagged polydT (for GAPDH internal control). Primers used were GGCCGTCATGGTGGCGAATGAAACCATAAAAAGTTGGAGGAAG, GCTAGCTTCAGCTAGGCATCAGTAGAAACAAGGAGTTT and CCAGA TCGTTCGAGTCGTTTTTTTTTTTTTTTTTTTTT for NA vRNA, cRNA and mRNA respectively, (tags underlined). Tagged cDNA was then diluted 1 in 10 and quantified using real-time quantitative PCR (qPCR) using Fast SYBR green master mix (Thermo Scientific). Primer pairs used were: CCTTCCCCTTTTCGATCTTG/ GGCCGTCATGGTGGCGAAT (NA vRNA), CTTTTTGTGGCGTGAATAGTG/ GCTAGCTTCAGCTAGGCATC (NA cRNA), CTTTTTGTGGCGTGAATAGTG/ CCAGATCGTTCGAGTCGT (NA mRNA) and AATCCCATCACCATCTTCCA/ TGGACTCCACGACGTACT CA (GAPDH). qPCR analysis was carried out in duplicate on a Viia 7 real-time PCR system (Thermo Fisher). Fold changes in gene expression relative to input (0 h.p.i.) were calculated using the $2^{-\Delta\Delta CT}$ method with GAPDH expression as internal control.

### Animal studies

Six- to eight-week-old female BALB/c (Envigo RMS UK Ltd) mice were maintained in pathogen-free conditions until used for viral infection. Mice were anesthetized using isoflurane and infected intranasally with $10^5$ PFU influenza virus in a 35 µl volume or sterile PBS (mock). Animals were monitored and weighed daily. Lungs were harvested on Day 2 and Day 7, or when weight loss dropped below 80% of the original weight on Day 0. Lungs were weighed, suspended in 1ml of PBS and homogenized using 2.8 mm beads and frozen at −80 °C.

### Reporting summary

Further information on research design is available in the Nature Portfolio Reporting Summary linked to this article.

## Data availability

The data generated in this study are provided in the Supplementary Information/Source Data file. The sequencing data used in this study are available in Genbank under accession numbers OR079445-OR079485. Protein structures PDB: 6XZP and PDB: 6QX8 were used to map mutations generated from this study. Source data are provided with this paper.

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

## Acknowledgements
This work was supported by grant 205100 from the Wellcome Trust (C.M.S., D.H.G., O.P., K.S. and W.B.) and by a Wellcome Trust Studentship (O.S.). Additional funding support from MRC grant MR/R009945/1 (E.S. and E.F.), an MRC DTP studentship (R.P.), BBSRC grant BB/R007292/1 (L.B.), Wellcome Trust grant 200187 (R.F.) and BBSRC grant BB/R013071/1 (T.P.P. and W.S.B.).

## Author contributions
Conceptualization: C.M.S., D.H.G., E.S., W.S.B. Funding acquisition: E.F., W.S.B. Investigation: C.M.S., D.H.G., O.S., E.S., R.P., O.K.P., K.S., L.B., R.F. Data curation: T.P.P. Supervision: C.M.S., E.F., W.S.B. Writing—original draft: C.M.S., D.H.G., O.S., R.P. Writing—review & editing: C.M.S., D.H.G., O.S., E.S., R.P., O.K.P., K.S., L.B., R.F., T.P.P., E.F., W.S.B.

## Competing interests
The authors declare no competing interests.
