## [Peer Review File · Nature Communications]

REVIEWER COMMENTS

Reviewer #1 (Remarks to the Author):

ANP32 proteins are essential cellular co-factors for the influenza virus polymerase. Species-specific differences in ANP32 proteins and adaptations in the polymerase impact viral host range. Human influenza polymerases have been shown to be dependent on human ANP32A and ANP32B. Here, Goldhill, Sheppard, et al. show that in the absence of ANP32A or B, the viral polymerase can evolve to utilize another family member, ANP32E. The authors focus on one evolved variant they identified with mutations in both PB1 and PA. They demonstrate that virus and polymerase encoding these mutations utilize ANP32E, and that elimination of all ANP32 proteins completely impairs polymerase function. Moreover, variants like these were previously identified as mouse-adaptive changes, and the authors show increased usage of murine ANP32B and increased pathogenesis in mice for their evolved virus. Mechanistic studies suggest a regulatory scheme where adaptive mutations alter the balance between formation of the symmetric polymerase dimer and the asymmetric polymerase dimer, which is bridged by ANP32. While the data are suggestive and this could be a satisfying model, many important controls are missing and mechanistic studies rely exclusively on artificial over-expression.

1. All interaction studies rely on over-expression systems, often in eHAP cells. Confirming some of these data in infected lung cells is important to establish their biological relevance.
2. Important controls are missing throughout. Polymerase activity and interaction studies utilize luciferase as a readout. However, control blots are missing in many of these assays. It is important to show that changes in luciferase are due to the activity under investigation, and not simply due to changes in protein expression. This includes experiments in Fig 2A, 2C, 3B, 3D, 4D and 5C. In addition, the co-immunoprecipitation assays in Fig 4B lacks a negative control, making it challenging to differentiate non-functional interactions with the polymerase from background binding.
3. Fig 3D utilizes triple KO cells to suggest ANP32 proteins are essential for replication. These experiments still show infectious virus that can be recovered from the triple KO cells, which the authors suggest are residual input virus. To make the claim the virus cannot replicate without ANP32 proteins, it is important to provide additional data to determine the source of this infectious virus.
4. A Gluc-complementation assay is used to measure polymerase dimer formation. The authors claim it selectively measures the symmetric dimer. However, this claim is not supported by current data; the increase in complementation upon addition of huANP32B suggest that at least some of the Gluc complementation comes from the asymmetric dimer. It would be important to validate the experimental system, perhaps using mutants known to exhibit defects in symmetric and asymmetric dimerization.

Minor:

1. Fig 1D present a growth curve. However, viral titers have already reached a peak plateau at the first time point investigated, preventing any assessment of replication kinetics.
2. Line 509 says infections are assessing viral fitness, but the experimental setup cannot assess fitness, rather just pathogenicity and replication in mice.

Reviewer #2 (Remarks to the Author):

huANP32A and huANP32B are required for influenza virus replication. Therefore, huANP32A/B knockout cells (DKO cells) cannot support virus replication. However, the authors found that mutant viruses that evolved to utilize huANP32E emerged after 2 passages in DKO cells. PB1-K577E and PA-Q556R allow the polymerase to interact with ANP32E and alter the balance between the two alternate forms of the polymerase dimers. Furthermore, these mutations also allow the polymerase to use mouse ANP32 and mediate adaptation of influenza virus to mice.

The manuscript is well written and the results are clearly presented. But, I have some questions, as outlined below.

1. No legends for the supplementary figures are provided.
2. Do double knockout cells and triple knockout cells grow as well as control cells? I am concerned that the speed of cell proliferation has an impact on virus replication.
3. Figure 1D. The titers of WT virus and mutant virus in MDCK cells are almost the same. Are these differences really statistically significant?

Reviewer #3 (Remarks to the Author):

In this study, Goldhill et al. presented important findings on the experimental adaptation of an influenza A virus, switching the host receptor from huANP32A/B to huANP32E. The authors first generated a 6+2 reassortant virus, comprising two HA and NA segments from human H1N1 virus (A/Puerto Rico/8/1934) and six internal segments from highly pathogenic avian influenza (HPAI) A/H5N1 virus (A/turkey/Turkey/1/2005). They passaged the 6+2 reassortant virus in DKO cells which lacked human ANP32A and ANP32B. After 4 passages, this study identified dual mutations on the PB1 (K577E) and PA (Q556R) genes. These mutations were experimentally tested in vitro and in vivo, this study demonstrated that K557E+Q556R mutations can increase polymerase activity. The results are interesting, the K577E+Q556R mutations also allowed the reassortant virus to use an alternative host factor, which is ANP32E.

The manuscript is well-written, the experimental approaches used by the authors are nicely designed and complex techniques are used. This study is important because they can shed light on identifying crucial mutations on influenza polymerase genes that may affect host range.

Comments.

1. The authors might want to discuss why highly pathogenic polymerase internal genes are used? Any other avian influenza strains (LPAI or HPAI) have been looked at?

2. In line 87–90 (last paragraph of the Introduction), this was stated: “we investigated whether influenza virus could evolve to replicate in human cells lacking expression of ANP32A and ANP32B. We found that an avian H5N1 virus could indeed evolve to do this, and we demonstrate that the mutated influenza polymerase was able to co-opt an alternative host factor, ANP32E.” The studied virus is a 6+2 reassortant (with H1 and NI from human, and internal genes from avian), but not an avian H5N1 influenza virus that exist in the general population. It is still not known whether a wild type H5N1 virus can evolve and confer efficient replication using the same approach. So line 89 needs to re-phrase. In a similar context, the authors used “an influenza A virus” which is quite vague in the Abstract. This needs to be clearer, that is, a human influenza virus with avian polymerase complex.

3. In addition to the K577E and Q556R mutations, the authors observed mutations in other genes (as shown in Fig 1C). In particular, L201F mutation on PB2 has occurred in two of three populations. This mutation looks interesting, did the authors attempt to test this? Also, are there any HA and NA mutations or no mutation observed after 4 passages?

4. Please provide some detailed descriptions on the reverse genetics methods.

REVIEWER COMMENTS

Reviewer #1 (Remarks to the Author):

ANP32 proteins are essential cellular co-factors for the influenza virus polymerase. Species-specific differences in ANP32 proteins and adaptations in the polymerase impact viral host range. Human influenza polymerases have been shown to be dependent on human ANP32A and ANP32B. Here, Goldhill, Sheppard, et al. show that in the absence of ANP32A or B, the viral polymerase can evolve to utilize another family member, ANP32E. The authors focus on one evolved variant they identified with mutations in both PB1 and PA. They demonstrate that virus and polymerase encoding these mutations utilize ANP32E, and that elimination of all ANP32 proteins completely impairs polymerase function. Moreover, variants like these were previously identified as mouse-adaptive changes, and the authors show increased usage of murine ANP32B and increased pathogenesis in mice for their evolved virus. Mechanistic studies suggest a regulatory scheme where adaptive mutations alter the balance between formation of the symmetric polymerase dimer and the asymmetric polymerase dimer, which is bridged by ANP32. While the data are suggestive and this could be a satisfying model, many important controls are missing and mechanistic studies rely exclusively on artificial over-expression.

1. All interaction studies rely on over-expression systems, often in eHAP cells. Confirming some of these data in infected lung cells is important to establish their biological relevance.

Whilst we agree it would be interesting to address the role of ANP32E in more physiologically relevant cell lines, the generation of ANP32-double and triple knockout primary airway cell cultures is beyond the scope of this manuscript, requiring substantial genetic editing of primary cells. The eHAP cells we have used here are immortalized cancer cell lines, but we have used them extensively in our previous work to illustrate interactions between influenza virus polymerase and the host ANP32 proteins. There is no evidence that the ANP32 host factors behave differently in their support of influenza polymerase in for example, A549 and Calu 3 immortalized lung cells, than in the immortalized eHAP.

Even if we did generate primary airway cells altered in ANP32 genes, it is not possible to perform infections in such cells with the viruses we have used in this study because we purposefully employed recombinant viruses with the HA and NA from the laboratory adapted strain PR8. This strategy was adopted because the PR8 virus does not infect human airway cells nor people. The PR8 virus is in fact highly mouse adapted. The use of the laboratory adapted PR8 HA and NA on the recombinant viruses we chose did allow us to perform mouse infection experiments (Figure 7B and C) with the PB1-K577E + PA-Q556R mutant that demonstrate the biological outcome in lung cells in vivo that arises from increased use of otherwise suboptimal ANP32 proteins.

Finally, we point out that this manuscript is not necessarily suggesting that the use of ANP32E or other suboptimal ANP32 proteins is biologically relevant for wild type influenza viruses, but that the experimentally evolved viruses we generate and characterize here, that acquire the ability to interact with those factors in artificially manipulated systems, reveal insights into how the polymerase complex is regulated.

2. Important controls are missing throughout. Polymerase activity and interaction studies utilize luciferase as a readout. However, control blots are missing in many of these assays. It is important to

show that changes in luciferase are due to the activity under investigation, and not simply due to changes in protein expression. This includes experiments in Fig 2A, 2C, 3B, 3D, 4D and 5C.

We have now supplemented figures presenting polymerase assay and split-luciferase assays with Western blots to indicate protein expression levels from transfected plasmids. This includes immunoblotting for polymerase subunits PA, PB1, homologues of ANP32 and mutant/tagged versions thereof. We provide a new Figure 2A and 3B that shows enhanced expression and/or stability of the mutant PB1 but not PA protein. We test the effect of these higher protein levels by titrating PB1 WT to the same level as PB1 K577E and demonstrate that this alone is not responsible for increased polymerase activity (Supplemental Figure 2).

We provide further new Western blots to accompany Figures 2C, 4D and 5C that show equivalent levels of the PA and PB1 fusion proteins used in the split luciferase assays. Therefore we conclude that changes in polymerase activity or binding is not a result of protein expression or stability.

In addition, the co-immunoprecipitation assays in Fig 4B lacks a negative control, making it challenging to differentiate non-functional interactions with the polymerase from background binding.

We have now included the control lanes of the co-immunoprecipitation experiment shown in Figure 4B. Here you can see that we do detect non-specific binding with the GFP-FLAG control at the same level as binding to ANP32-FLAG proteins by wild type polymerase. This is a consistent result and should be used to represent the background level of binding, which is similar for all ANP32 homolog binding to WT polymerase. In contrast, the level of mutant polymerase that co-precipitates with huANP32B and huANP32E is substantially above background.

References to the Western Blots and controls for the co-immunoprecipitation assay have been updated in the text; lines 344-349, 395-397, 437-448, 457-462.

3. Fig 3D utilizes triple KO cells to suggest ANP32 proteins are essential for replication. These experiments still show infectious virus that can be recovered from the triple KO cells, which the authors suggest are residual input virus. To make the claim the virus cannot replicate without ANP32 proteins, it is important to provide additional data to determine the source of this infectious virus.

We have repeated this assay using an acid wash step after virus inoculation to remove all external virus at the start of the experiment. The new data is presented as Figure 3D in the modified manuscript. There is now no 'background' level of either virus in TKO cells. The relevant text is now modified in lines 408-418.

4. A Gluc-complementation assay is used to measure polymerase dimer formation. The authors claim it selectively measures the symmetric dimer. However, this claim is not supported by current data; the increase in complementation upon addition of huANP32B suggest that at least some of the Gluc complementation comes from the asymmetric dimer. It would be important to validate the experimental system, perhaps using mutants known to exhibit defects in symmetric and asymmetric dimerization.

To validate our experimental approach we have included the PA-352-6A mutant which was identified and experimentally validated as disrupting the symmetric dimer interface by Fan et al, 2019. Indeed in our split-luciferase dimerisation assay now presented in the modified Figure 5C, the

polymerases reconstituted with PA-352-6A do not exhibit significant binding above the –PA control. Western blot shows this is not attributed to a difference in protein expression.

We have now removed the data around the effect of overexpressed human ANP32B on the split luciferase signals we measure in this assay. We feel their interpretation is beyond the scope of this manuscript.

To further support our findings that suggested a change in polymerase dimerization conferred by the mutations, we have added new in vitro data (Figure 5D). We have purified both WT and mutant Tky05 polymerase from Sf9 insect cells and performed size exclusion chromatography experiments. In comparison to Tky05 WT, the elution profile of PB1-K577E + PA-Q556R corresponds to that of monomeric polymerase. Under these experimental conditions (ie. in the absence of ANP32) only the symmetric dimer structure has been observed (Fan et al, 2019, Chang et al, 2015). This is further evidence that symmetric dimer formation is abrogated by PB1 K577E + PA Q556R.

Text referring to the new data has been added to lines 499-505.

Minor:

1. Fig 1D present a growth curve. However, viral titers have already reached a peak plateau at the first time point investigated, preventing any assessment of replication kinetics.

We agree with the reviewer and have consequently repeated the growth curves by including earlier time points - 8, 16, 24, 48 and 72 hrs post infection (Figures 1D & E). There is a marginal difference in the growth kinetics of the WT and PB1-K577E + PA-Q556R viruses in MDCK cells and slightly accelerated growth of the PB1-K577E + PA-Q556R virus in the eHAP1-DKO cells at 8 hrs post infection, however, this does not result in an overall growth advantage as the time course progresses. As expected the WT virus fails to grow in DKO cells until post 50 hrs - likely evidence of evolved virus as observed in Figure 1A.

These new data replace the previous growth curve in Figure 1D and E and are described in lines 317-326.

2. Line 509 says infections are assessing viral fitness, but the experimental setup cannot assess fitness, rather just pathogenicity and replication in mice.

The terminology has now been corrected in the text, line 544.

Reviewer #2 (Remarks to the Author):

huANP32A and huANP32B are required for influenza virus replication. Therefore, huANP32A/B knockout cells (DKO cells) cannot support virus replication. However, the authors found that mutant viruses that evolved to utilize huANP32E emerged after 2 passages in DKO cells. PB1-K577E and PA-Q556R allow the polymerase to interact with ANP32E and alter the balance between the two alternate forms of the polymerase dimers. Furthermore, these mutations also allow the polymerase to use mouse ANP32 and mediate adaptation of influenza virus to mice.

The manuscript is well written and the results are clearly presented. But, I have some questions, as

outlined below.

1. No legends for the supplementary figures are provided.

We apologise for this oversight, the supplementary figure legends have now been included, lines 797-817.

2. Do double knockout cells and triple knockout cells grow as well as control cells? I am concerned that the speed of cell proliferation has an impact on virus replication.

Surprisingly given the importance of ANP32 proteins in a multitude of cellular processes, we do not observe any profound differences in their growth over passage, so we do not believe that slower growth can explain the lack of virus replication. Indeed, the DKO cells supported peak titres of virus (mutant) around 10^6 pfu/ml equivalent to those from WT cells (Figures 1B, 1E and 3C). Moreover, in Figure 3D we demonstrate that viral growth can be rescued back even in the TKO cell line by exogenous expression of ANP32 proteins, although admittedly the peak titres reached are only around 10^5 pfu/ml. However one should note that since we are only complementing ANP32 using plasmid transfection not all cells will be capable of supporting replication in this assay.

3. Figure 1D. The titers of WT virus and mutant virus in MDCK cells are almost the same. Are these differences really statistically significant?

Following the recommendations from Reviewer 1 we have now repeated these growth curves with a broader range of time points (Figures 1D & E). Our conclusion from these experiments is that there are marginal differences between WT and mutant virus in MDCK cells and in DKO cells. The new figures and accompanying text are now incorporated in the modified manuscript lines 317-326.

Reviewer #3 (Remarks to the Author):

In this study, Goldhill et al. presented important findings on the experimental adaptation of an influenza A virus, switching the host receptor from huANP32A/B to huANP32E. The authors first generated a 6+2 reassortant virus, comprising two HA and NA segments from human H1N1 virus (A/Puerto Rico/8/1934) and six internal segments from highly pathogenic avian influenza (HPAI) A/H5N1 virus (A/turkey/Turkey/1/2005). They passaged the 6+2 reassortant virus in DKO cells which lacked human ANP32A and ANP32B. After 4 passages, this study identified dual mutations on the PB1 (K577E) and PA (Q556R) genes. These mutations were experimentally tested in vitro and in vivo, this study demonstrated that K577E+Q556R mutations can increase polymerase activity. The results are interesting, the K577E+Q556R mutations also allowed the reassortant virus to use an alternative host factor, which is ANP32E.

The manuscript is well-written, the experimental approaches used by the authors are nicely designed and complex techniques are used. This study is important because they can shed light on identifying crucial mutations on influenza polymerase genes that may affect host range.

Comments.

1. The authors might want to discuss why highly pathogenic polymerase internal genes are used? Any other avian influenza strains (LPAI or HPAI) have been looked at?

This study was conceptualised based on earlier observations of low levels of replication of the Tky05 (6:2 reassortant) virus in the eHAP1-DKO cells. Comparable experiments conducted with A/Victoria/3/75 (H3N2), A/turkey/England/50-92/91 (H5N1), PB2-6267K, A/England/195/2009 (pH1N1) and A/PR/8/34 (H1N1) showed no detectable growth in DKO cells at 72 hrs post infection (Staller et al., 2019). Therefore the only scientific option was to use the virus with these internal genes from the highly pathogenic avian influenza virus. We have now expanded this explanation in the methods and results; text lines 111-118 and 296-303.

2. In line 87–90 (last paragraph of the Introduction), this was stated: “we investigated whether influenza virus could evolve to replicate in human cells lacking expression of ANP32A and ANP32B. We found that an avian H5N1 virus could indeed evolve to do this, and we demonstrate that the mutated influenza polymerase was able to co-opt an alternative host factor, ANP32E.” The studied virus is a 6+2 reassortant (with H1 and NI from human, and internal genes from avian), but not an avian H5N1 influenza virus that exist in the general population. It is still not known whether a wild type H5N1 virus can evolve and confer efficient replication using the same approach. So line 89 needs to re-phrase. In a similar context, the authors used “an influenza A virus” which is quite vague in the Abstract. This needs to be clearer, that is, a human influenza virus with avian polymerase complex.

We have reworded the line of the introduction to which the reviewer refers as follows:

‘We describe one example of a virus that showed limited replication in cells lacking human ANP32A and B, and evolved in those cells to increase replication efficiency compared to the ancestor virus ‘.

In addition as stated above, we have expanded the section in results that explains the origins of gene segments of the virus we have used here and why we chose this one; text lines 111-118 and 296-303.

3. In addition to the K577E and Q556R mutations, the authors observed mutations in other genes (as shown in Fig 1C). In particular, L201F mutation on PB2 has occurred in two of three populations. This mutation looks interesting, did the authors attempt to test this? Also, are there any HA and NA mutations or no mutation observed after 4 passages?

We agree that the L201F mutation in PB2 is of interest as it localises close to the N-terminal face of ANP32 and therefore may also serve to enhance an interaction with huANP32E. However, our preliminary experiments with L201F did not reveal any encouraging results under our experimental conditions. These additional mutations are the subject of ongoing investigations.

We have sequenced genome segments of the evolved mutant virus after 2 and 4 passages and confirmed that the HA and NA genes do not possess any mutations. We have now added this information to the text lines 314-317.

4. Please provide some detailed descriptions on the reverse genetics methods.

We did not generate any new viruses for this manuscript specifically using reverse genetics. We now refer to our previous publication that described the methods used to generate our starting virus, Li et al PLoSPath 2018 in text lines 111-118 and 296-303.

REVIEWER COMMENTS

Reviewer #1 (Remarks to the Author):

The authors have provided additional control data that have strengthened their conclusions. The blots and negative control pulldowns established experimental variance, but the authors provide additional data or context to eliminate these as contributing to the main results. The “acid wash” step in Fig 3D allows for very clear demonstration that these cells do not support replication. Validation of the symmetric vs asymmetric dimer Gluc assay provides confidence for their data, and also a useful experimental system for the field. The attention to detail and full descriptions of these changes are greatly appreciated.

The authors describe new replication with earlier time points, referring to revised panels Fig 1D-E. However, the resubmission contains the original Fig 1. If the new data are as described, they address the original concern.

Labelling in Fig 5C (PA 324-326A) does not match the PA dimer mutant described on line 489 (PA 352-356, following Fan 2019).

During the original review, it was noted that all *in vitro* interaction studies were done in over-expression systems, and it was requested that key results be confirmed under more physiological settings. In their response, the authors either misunderstood, for which this reviewer apologizes for not being clear enough, or misinterpreted this critique. The request was not to use primary lung cells, nor did it ask for primary viral isolates. Instead, it was to verify protein:protein interactions under conditions of infection, not artificial over-expression. Relying exclusively on over-expression can lead to spurious interaction and artifactual results. At a minimum, this caveat should be addressed in the text. As the reviewers rightly note, the use of ANP32E is likely not biologically relevant given the dominance of ANP32A and ANP32B. But, the validity of the interactions they describe is important for the functional model they put forward that helps to explain why ANP32 proteins are needed, suggesting they regulate the balance between symmetric and asymmetric dimers.

Reviewer #3 (Remarks to the Author):

The comments are well addressed.

REVIEWER COMMENTS

Reviewer #1 (Remarks to the Author):

The authors have provided additional control data that have strengthened their conclusions. The blots and negative control pulldowns established experimental variance, but the authors provide additional data or context to eliminate these as contributing to the main results. The “acid wash” step in Fig 3D allows for very clear demonstration that these cells do not support replication. Validation of the symmetric vs asymmetric dimer Gluc assay provides confidence for their data, and also a useful experimental system for the field. The attention to detail and full descriptions of these changes are greatly appreciated.

The authors describe new replication with earlier time points, referring to revised panels Fig 1D-E. However, the resubmission contains the original Fig 1. If the new data are as described, they address the original concern.

Apologies for the confusion, we did indeed include the original Fig 1 panel, however the text does refer to the new data figures which are now included.

Labelling in Fig 5C (PA 324-326A) does not match the PA dimer mutant described on line 489 (PA 352-356, following Fan 2019).

We thank the reviewer for noticing this inconsistency. The labelling in the figure has now been rectified to correctly match the text (and Fan et al, 2019).

During the original review, it was noted that all *interaction studies* were done in over-expression systems, and it was requested that key results be confirmed under more physiological settings. In their response, the authors either misunderstood, for which this reviewer apologizes for not being clear enough, or misinterpreted this critique. The request was not to use primary lung cells, nor did it ask for primary viral isolates. Instead, it was to verify protein:protein interactions under conditions of infection, not artificial over-expression. Relying exclusively on over-expression can lead to spurious interaction and artifactual results. At a minimum, this caveat should be addressed in the text. As the reviewers rightly note, the use of ANP32E is likely not biologically relevant given the dominance of ANP32A and ANP32B. But, the validity of the interactions they describe is important for the functional model they put forward that helps to explain why ANP32 proteins are needed, suggesting they regulate the balance between symmetric and asymmetric dimers.

Thank you for the clarification. We have added this caveat as a discussion point (lines 627-633). The later half of this paragraph now reads:

“Our observations could be explained by the escape mutations altering a balance between the two alternate forms of polymerase dimers (Figure 8). By reducing the ability of the polymerase to form the symmetric dimer, the virus favours the formation of the asymmetric dimer even with a suboptimal ANP32. Thus the balance between monomeric (transcriptionally competent), asymmetric and symmetric dimers is heavily influenced by the relative stoichiometry of FluPol and ANP32 proteins. One caveat to this study is that our interaction assays rely on overexpressed levels of protein that may not reflect the levels of endogenous ANP32 or changing FluPol levels throughout infection. More research will be needed to fully understand the balance of polymerase dimers and how this impacts on polymerase mode of action.”

Reviewer #3 (Remarks to the Author):

The comments are well addressed.